# Calibrated Preference Learning: The Case of Label Ranking

**Santo M. A. R. Thies** [1 2 3]  **Viktor Bengs** [1]  **Timo Kaufmann** [2 3]  **Sebastian J. Vollmer** [1]  **Eyke Hüllermeier** [1 2 3]

## Abstract

Calibration, the alignment of predicted probabilities with true outcome frequencies, is essential for reliable decision-making. While extensively studied for classification and regression, calibration has not been formally addressed for probabilistic label ranking, where the goal is to predict a distribution over orderings of a label set. Naively treating rankings as classes ignores their structure and fails to capture important modalities such as pairwise and top-k predictions. We formalize calibration for label ranking and develop a hierarchy of notions covering full rankings, sub-rankings, and top-k rankings. We prove that full-rank calibration implies the others but not conversely, and sub-ranking and top-k calibration are incomparable. Empirically, we find popular label ranking models are often poorly calibrated, with substantial differences between sub-ranking and top-k metrics. Applying our framework to RLHF reward models, we find that calibration correlates strongly but not perfectly with benchmark accuracy, suggesting it captures a meaningful quality dimension beyond top-1 accuracy. These findings motivate future work on understanding the downstream effects of miscalibration and developing methods to correct it.

## 1. Introduction

Probabilistic label ranking (ProLR) models a probability distribution over the possible rankings of a set of items given a context, capturing both the probability and the certainty of each potential ranking (Cheng et al., 2010; Cheng & Hüllermeier, 2009). For such predictions to be trustwor-

[1]Data Science and its Applications, Deutsches Forschungsinstitut für Künstliche Intelligenz (DFKI), Kaiserslautern, Germany [2]Munich Center for Machine Learning, Munich (MCML), Germany [3]Ludwig Maximilian Universität München, Munich, Germany. Correspondence to: Santo, Thies <santo.thies@kiml.ifi.lmu.de>.

*Proceedings of the 43rd International Conference on Machine Learning*, Seoul, South Korea. PMLR 306, 2026. Copyright 2026 by the author(s).

thy, the model must be *calibrated*: predicted probabilities should align with true outcome frequencies (Silva Filho et al., 2023). While calibration has been extensively studied in classification (Vaicenavicius et al., 2019), regression (Song et al., 2019), and recommender systems (da Silva & Jannach, 2026), calibration for distributions over rankings remains unexplored. One important application is Reinforcement Learning from Human Feedback (RLHF) (Ouyang et al., 2022), which aims to align LLMs to human preferences via reward models that learn underlying preference structures. The alignment procedure in RLHF relies on pairwise preferences, which are substructures of full rankings as considered in label ranking (Wirth et al., 2017).

Label ranking aims to predict a complete ordering of a set of items given some context (Fürnkranz et al., 2008; Hüllermeier et al., 2008). Unlike standard label ranking, which predicts only the single most likely ordering (Vembu & Gärtner, 2010), ProLR exposes the distribution over rankings. This information, in turn, is only reliable for downstream decisions if the model is well calibrated (Silva Filho et al., 2023). In particular, calibration ensures that expressed uncertainty reflects true outcome variability, which is desirable for pluralistic alignment where consensus and contestation should be distinguished.

In ranking-related settings such as recommender systems (da Silva & Jannach, 2026), calibration has been viewed as aligning predicted item scores with underlying user scores. These score-based calibration methods (Yan et al., 2022; Sculley, 2010) then aim at aligning scores to correctly reflect how likely a user is in favor of an item to be ranked. A different line of work defines calibration via a pivot point that separates relevant from non-relevant items within a ranking (Fürnkranz et al., 2008). In contrast, aligning a predicted distribution to the underlying distribution over rankings, as required in ProLR, remains unexplored.

One can easily obtain a notion of calibration by treating the label ranking problem as a multi-class classification problem, with each ranking considered a class. However, the resulting notion of calibration has some disadvantages for practical applications. First, there is a factorial number of (ranking) classes, which makes the quantitative measurement of this kind of calibration computationally difficult even for a moderate number of labels. What complicates

matters further is that, in general, the rankings observed reflect only a tiny fraction of all possible outcomes. Furthermore, the metric structure of the ranking space is not reflected in the straightforward calibration notion for multi-class classification. For example, the rankings $i_1 \succ i_2 \succ i_3$ and $i_1 \succ i_3 \succ i_2$ both share the common sub-ranking $i_1 \succ i_2$, and, as such, calibration on the former may induce calibration on the latter (see Section 4), which remains unreflected in the naive classification viewpoint. Finally, important modalities for practical applications, including pairwise and top-k rankings, are not taken into account.

We therefore introduce the notion of calibration in ProLR, capturing calibration at different granularities, ranging from calibration over the full item set to calibration over selected subsets. Building upon the two forms of strong and weak calibration in multi-class classification, we develop a hierarchy of calibration notions for ProLR. We theoretically investigate the relationships between these notions, showing that calibration on sub-items does not imply calibration on all items.

**Contribution.**   Our main contributions are as follows:

1. **Calibration in Label Ranking:** We introduce calibration notions tailored to probabilistic label ranking, extending beyond what can be captured by viewing label ranking as a multi-class classification problem.

2. **Unified theoretical picture (Figure 1 and Figure 4):** We establish theoretical relationships between the proposed calibration notions, showing that calibration on sub-rankings does not imply calibration on full-rankings, also for widely used Plackett–Luce and Mallows models.

3. **Empirical Investigation:** We empirically evaluate the calibration properties of popular label ranking learners, illustrating differences between sub-ranking and full-ranking calibration. We also examine calibration of reward models in a popular RLHF benchmark.

## 2. Related Work

**Probabilistic Calibration**   Prior work on multi-class calibration has established a hierarchy of increasingly strong notions, ranging from calibration of only the most likely class (Guo et al., 2017), through *class calibration* (Zadrozny & Elkan, 2001), which decomposes into one-vs-rest binary tasks, to *multi-class calibration* (Widmann et al., 2019), which requires calibrated distributions over all classes. This hierarchy has been extended to the case of regression (Song et al., 2019; Widmann et al., 2021) and set-based probabilistic classifiers (Mortier et al., 2023; Jürgens et al., 2025). Our work introduces an analogous hierarchy of calibration

notions for the label ranking setting, additionally considering special cases such as pairwise and top-k calibration. Calibration has been investigated for popular machine learning algorithms, such as decision trees (Zadrozny & Elkan, 2001), random forests (Shaker & Hüllermeier, 2025) and neural networks (Mukhoti et al., 2020; Wang, 2023). In line with this, we investigate calibration of popular ranking models such as Plackett–Luce and Mallows models, showing that they are often poorly calibrated for large item sets. For an extensive overview of calibration methods, we refer to the reviews by Silva Filho et al. (2023) and Lane (2026).

**Label Ranking and Calibration.**   The field of preference learning has contributed significantly to the rise of LLMs, particularly to their fine-tuning (Ouyang et al., 2022; Kaufmann et al., 2025b). This development was accompanied by new research impulses in subfields of preference learning, such as label ranking. New learning methods (Korba et al., 2018; Adam et al., 2024; Thies et al., 2024; Zhou et al., 2024) and extensions of the standard setting have been introduced, such as the partial label ranking (Alfaro et al., 2021; 2023b) or dyad ranking (Schäfer & Hüllermeier, 2018) problem. Further work has improved inference for probabilistic ranking models, such as the popular Mallow's model (Kenig et al., 2018; Ping et al., 2020). Yet surprisingly, the concept of calibration has been completely neglected so far in label ranking, even though (i) calibration is of special interest to the emerging topic of uncertainty quantification in ML (Hüllermeier & Waegeman, 2021) and (ii) a well-calibrated label ranking model could be a useful tool to capture annotator differences and train models that reflect these nuances.

**Calibration in other Preference Learning Settings**   In contrast to ProLR, calibration in other ranking-based domains, particularly recommender systems, has received considerable interest. It should be noted that, in this domain, calibration refers to aligning predicted scores with the user's underlying latent scores. In ProLR, however, we consider aligning probability estimates to the true probabilities of the data-generating process. Sculley (2010) were the first to consider calibrated scores alongside accurate rank predictions. Li et al. (2015); Steck (2018); Penha & Hauff (2021) show that solely focusing on prediction quality in recommender systems deteriorates calibration, resulting in reduced user satisfaction. Calibration of top-k rankings based on recommender systems has been considered by Sato (2024). Similarly, Yan et al. (2022); Zhang et al. (2024) deal with increasing user satisfaction in recommender systems via a two-component loss function, where the latter part encourages calibration. To improve calibration, non-parametric (Menon et al., 2012) and parametric methods (Kweon et al., 2022) have been introduced, generalizations of classification methods. We refer to da Silva & Jannach (2026) for a broader overview.

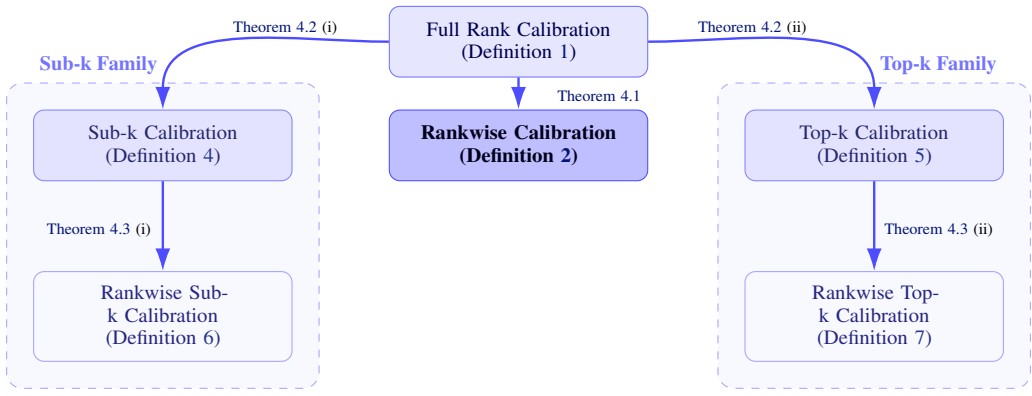

*Figure 1.* Overview of the calibration definitions and their relationships (an outgoing arrow means implication, whereas dashed only hold for specific model classes). Exclusion results are depicted in Figure 4.

**Calibration in RLHF.** The term calibration appears in multiple senses within the RLHF literature. One line of research studies calibration of LLM output confidence: whether models express well-calibrated uncertainty in their answers (Zhu et al., 2023; Tian et al., 2023; Stangel et al., 2025). A related question for our work is whether reward models themselves are calibrated. Halpern et al. (2025) formalize pairwise calibration for pluralistic alignment: they train ensembles of reward functions such that the ensemble prediction is calibrated with respect to the empirical annotator distribution. Kaufmann et al. (2025a) focuses on training a single reward model that captures utility differences well, thereby producing better-calibrated Bradley-Terry probabilities. Our work is more closely aligned with the second direction, as we focus on calibrating reward models themselves. It differs from prior work by formalizing calibration *notions* for ranking distributions, offering a framework for measuring and comparing calibration across prediction granularities and beyond RLHF, rather than proposing training methods.

## 3. Probabilistic Label Ranking

Here, we introduce the general notation of probabilistic label ranking and formalize the learning problem. We then briefly review common probabilistic ranking models, including the Plackett–Luce and Mallows models. Finally, we discuss the connection between the Plackett–Luce model and the widely adopted learning paradigm of ranking by pairwise comparison. For a more general overview of label ranking, see (Vembu & Gärtner, 2010; Zhou et al., 2014).

**Notation.** Let $\mathcal{X}$ denote the feature space, $\mathcal{I}$ the set of items, and $\mathcal{B} \subseteq \mathcal{I}$ a subset of items. The set $\mathcal{S}_\mathcal{I}$ consists of all rankings of $\mathcal{I}$, with $m = |\mathcal{I}|$ denoting the number of items. A ranking $\pi \in \mathcal{S}_\mathcal{I}$ is a mapping $\pi : \mathcal{I} \to \mathbb{N}$, which assigns each item its corresponding position in the ranking. We write a ranking as $\pi = i \succ j \succ k$ instead

of $\pi(i) = 1, \pi(j) = 2, \pi(k) = 3$. We also use the $\succ$ operator to represent a ranking $\pi$ through (sub-)rankings $\pi_1, \pi_2$, e.g., $\pi = \pi_1 \succ \pi_2$. For each ranking $\pi$, we denote by $\pi^{-1} : \mathbb{N} \to \mathcal{I}$ the mapping assigning each position its corresponding item, e.g., if $\pi = i_1 \succ i_3 \succ i_2$, then $\pi^{-1}(1) = i_1, \pi^{-1}(2) = i_3$, and $\pi^{-1}(3) = i_2$. Furthermore, given $\rho \in \mathcal{S}_\mathcal{B}$ with $\mathcal{B} \subseteq \mathcal{I}$, we write $\rho \subseteq_\mathcal{B} \pi$ if $\forall i, j \in \mathcal{B} : \rho(i) < \rho(j) \Leftrightarrow \pi(i) < \pi(j)$ and omit $\mathcal{B}$ when it is clear from context. We denote random variables by capital letters $X, \Pi$, sets by calligraphic or script letters such as $\mathcal{X}, \mathcal{S}_\mathcal{I}$, and realizations by lowercase letters $x, \pi$. The probability of an event $A$ is denoted by $P(A)$, while $\mathbb{P}(\mathcal{A})$ denotes the set of all probability distributions over a set $\mathcal{A}$.

**Probabilistic Label Ranking.** Let a hypothesis space $\mathcal{H}$ be given, where a model $h \in \mathcal{H}$ is a mapping $h : \mathcal{X} \to \mathbb{P}(\mathcal{S}_\mathcal{I})$. For a ranking $\pi \in \mathcal{S}_\mathcal{I}$ and a given context $x \in \mathcal{X}$, $h(x)$ denotes a distribution over $\mathcal{S}_\mathcal{I}$, and $h(x)[\pi]$ denotes the probability assigned to $\pi$. A model $h$ is learned by minimizing a chosen loss function on a dataset $\mathcal{D} = \{(x_i, \pi_i)\}_{i=1}^n$ with $x_i \in \mathcal{X}$ and $\pi_i \in \mathcal{S}_\mathcal{I}$. Throughout this work, the data points are assumed to be independent and identically distributed. Similarly to standard supervised learning, a common approach in probabilistic label ranking is to induce a parametrized ranking model of the form $h(x) = f(g(x))$, where $f : \Theta \to \mathbb{P}(\mathcal{S}_\mathcal{I})$ is a parametrized ranking distribution and $g : \mathcal{X} \to \Theta$ predicts the per-instance parameters. Since $f$ is fixed once $\theta \in \Theta$ is given, learning reduces to finding the optimal function $g$.

**Plackett–Luce Model.** A prominent class of models for $f$ is latent utility models, which are parametrized by $\theta \in \Theta \subset \mathbb{R}_+^m$. Widely adopted is the Plackett–Luce (PL) model, which assigns a probability to a ranking $\pi \in \mathcal{S}_\mathcal{I}$ via

$$q^{\boldsymbol{\theta}}[\pi] = \prod_{i=1}^m \frac{\boldsymbol{\theta}_{\pi^{-1}(i)}}{\sum_{j=i}^m \boldsymbol{\theta}_{\pi^{-1}(j)}} . \tag{1}$$

The parameters are typically learned by minimizing the negative log-likelihood of the PL model on the training data.

As the PL model is identifiable only up to a positive scaling factor (Alvo & Philip, 2014), the parameters are commonly constrained such that $\sum_{i=1}^{m} \theta_i = 1$ for all $x \in \mathcal{X}$ [1]. We denote the resulting hypothesis space by $\mathcal{H}_{PL}$. For two-item rankings, i.e., pairwise comparisons, PL corresponds to the Bradley–Terry (Bradley & Terry, 1952) model (see Hunter (2004)) defined as $P(i \succ j) = \frac{\theta_i}{\theta_i + \theta_j}$.

**Mallows Model.** In contrast to latent utility models, another popular class of models for $f$ is distance-based models. The most prominent example is the Mallows model (Mallows, 1957), parametrized by $(\tau, \lambda) \in \Theta = \mathcal{S}_\mathcal{I} \times \mathbb{R}_+$. The probability of a ranking $\pi$ under the Mallows model $(\tau, \lambda)$ is

$$q^{(\tau, \lambda)}[\pi] = \tfrac{1}{C(\lambda, \tau)} \exp(-\lambda d(\pi, \tau)), \qquad (2)$$

where $C(\lambda, \tau) = \sum_{r \in \mathcal{S}_\mathcal{I}} \exp(-\lambda d(r, \tau))$ is a normalization constant and $d : \mathcal{S}_\mathcal{I} \times \mathcal{S}_\mathcal{I} \to \mathbb{R}$ is a distance defined on rankings. In practice, both parameters are learned by alternating between optimizing $\lambda$ and updating $\tau$ until convergence (Busse et al., 2007). Throughout this work, we use the Kendall distance for $d$ (see Alvo & Philip, 2014) and denote the resulting hypothesis space by $\mathcal{H}_{MM}$.

**Ranking by Pairwise Comparison.** Finally, we consider the approach of *ranking by pairwise comparison* (RPC) (Hüllermeier & Furnkranz, 2004). In this approach, a ranking model of the form $h(x) = f(g(x))$ is learned, where $f : [0,1]^{m \times m} \to \mathbb{P}(\mathcal{S}_\mathcal{I})$ is a mapping from pairwise preference probabilities to a predictive ranking distribution and $g : \mathcal{X} \to [0,1]^{m \times m}$ predicts the pairwise preference probabilities for each instance $x \in \mathcal{X}$. The idea is (i) to learn pairwise preference probabilities via $g$ for every item pair by reducing the label ranking problem to $\binom{m}{2}$ binary classification problems – one for each pair, and (ii) to aggregate these pairwise preference probabilities to a predictive ranking (distribution) via $f$. For each item pair $(i,j)$, the dataset $\mathcal{D}_{i,j} = \{(x, \mathbb{1}_{\pi \supseteq i \succ j}) \mid (x, \pi) \in \mathcal{D}\}$ is considered for training a binary classifier $g_{i,j} : \mathcal{X} \to [0,1]$. This leads to a pairwise probability matrix $\mathbf{M}(x) = [g_{i,j}(x)]_{i,j \in \mathcal{I}}$, with $g_{j,i}(x) = 1 - g_{i,j}(x)$. While there are several options available for $f$ (see Fahandar et al., 2017), they often yield degenerate predictive distributions. In ProLR, aggregations for $f$ are of interest that can produce non-degenerate predictive distributions. One way to achieve this is by interpreting the entries of $\mathbf{M}(x)$ as Bradley–Terry probabilities and recovering Plackett–Luce weights $\theta_1, \ldots, \theta_m$ such that $\frac{\theta_i}{\theta_i + \theta_j} \approx g_{i,j}(x)$, e.g., by using the improved Zermelo algorithm (Newman, 2023).

---

[1] For stability, $g$ often outputs log scores: $\theta_i = \exp(g(x)_i)$.

## 4. Calibration in ProLR

To introduce meaningful calibration concepts in label ranking, we follow the very principle of calibration. We will briefly examine this idea in its two common variants for multi-class classification, then apply it to label ranking and adapt it to pairwise and top-k rankings. All proofs can be found in Appendix A.

**Calibration in Multi-class Classification** In multi-class classification, we are given a finite set of outcomes $\mathcal{Y}$, for which a model $h : \mathcal{X} \to \mathbb{P}(\mathcal{Y})$ is learned. The outcomes are distributed according to some random variable $Y$. We call a model *strongly calibrated* if it holds for all $y \in \mathcal{Y}$ that:

$$\forall p \in \mathbb{P}(\mathcal{Y}) : P(Y = y \mid h(X) = p) = p[y]. \qquad (3)$$

In words, if a model $h$ is strongly calibrated, the probability of class $y$ given the predicted distribution $h(X) = p$ is exactly $p[y]$. For example, in weather forecasting with $\mathcal{Y} = \{\text{sun}, \text{rain}, \text{snow}\}$, strong calibration requires that among all days assigned prediction $(0.5, 0.3, 0.2)$, outcome frequencies match it.

One can weaken this notion naturally by restricting the condition to the respective entry of $h(X)$. Formally, a classifier is called *weakly calibrated* if it holds for all $y \in \mathcal{Y}$ that

$$\forall \alpha \in [0,1] : P(Y = y \mid h(X)[y] = \alpha) = \alpha. \qquad (4)$$

In the weather example, among days where rain is predicted with probability $\alpha$, it rains on an $\alpha$-fraction (and analogously for sun and snow). These can diverge because calibration pools by the outcome's probability: predictions $(0.6, 0.3, 0.1)$ and $(0.4, 0.3, 0.3)$ (perhaps reflecting different seasonal patterns) both assign $0.3$ to rain, so weak calibration pools them. If rain frequencies are $0.2$ and $0.4$ respectively, they average to $0.3$ (weak holds), but neither full prediction matches its true distribution (strong fails).

**Calibration for Full Rankings in ProLR** Translating the concepts to label ranking can be easily done by viewing the problem as a multi-class classification problem, where each ranking is interpreted as a class. Thus, for an item set $\mathcal{I}$ with $|\mathcal{I}| = m$, there exist $m!$ different rankings, which correspond to $m!$ different classes. With this, we introduce the notion of full-rank calibration (Definition 1), which is the label ranking equivalent to strong calibration in the sense of multi-class classification.

**Definition 1.** A model $h \in \mathcal{H}$ is *full-rank calibrated* if and only if it holds for all $\pi \in \mathcal{S}_\mathcal{I}$ that

$$\forall p \in \mathbb{P}(\mathcal{S}_\mathcal{I}) : P(\Pi = \pi \mid h(X) = p) = p[\pi]. \qquad (5)$$

In a similar fashion, we introduce the notion of *rankwise calibrated* (Definition 2), which considers conditioning only

on the probability of a specific ranking, instead of a full probability distribution as in full-rank calibration.

**Definition 2.** A model $h \in \mathcal{H}$ is *rankwise* calibrated if and only if it holds for all $\pi \in \mathcal{S}_\mathcal{I}$ that

$$\forall \alpha \in [0, 1] : \ P(\Pi = \pi \mid h(X)[\pi] = \alpha) = \alpha. \quad (6)$$

As an example, consider predicting musical taste with items *rock*, *pop*, and *classical*. Rankwise calibration requires that among all people for whom a ranking like rock $\succ$ pop $\succ$ classical is predicted with probability $\alpha$, it occurs for an $\alpha$-fraction. Full-rank calibration requires that among all people assigned the same distribution over all six rankings, empirical frequencies match it. As with weather, these can diverge because rankwise calibration pools by one ranking's probability: different full predictions (e.g., different age groups) assigning the same probability to rock $\succ$ pop $\succ$ classical are pooled, allowing subgroups to average correctly while neither matches the full prediction. One can easily show that full-rank calibration implies rankwise calibration, as formulated in Theorem 4.1.

**Theorem 4.1.** *For an item set $\mathcal{I}$ and rankings $\mathcal{S}_\mathcal{I}$, it holds*

$$h \text{ is full-rank calibrated} \ \Rightarrow h \text{ is rankwise calibrated.}$$

Intuitively, a model can be rankwise calibrated yet not full-rank calibrated, as the latter is a more demanding property. As Theorem A.1 shows, this intuition holds under some mild assumptions satisfied by PL and Mallows models.

**Calibration for Key Modalities in ProLR.** As noted in the introduction, the classification viewpoint ignores relationships between rankings and important modalities such as pairwise and top-k rankings. To address this, we introduce calibration notions for sub-rankings and top-k rankings, projecting full rankings onto the sub-rankings of interest and marginalizing the resulting distributions accordingly.

Consider a subset of items $\mathcal{B} \subseteq \mathcal{I}$ with $|\mathcal{B}| \geq 2$ for which $\rho \in \mathcal{S}_\mathcal{B}$ is a partial ranking of interest. Then we denote by

$$\mathcal{P}_\mathcal{B}(\rho) = \{\pi \in \mathcal{S}_\mathcal{I} \mid \pi \supseteq \rho\}, \quad (7)$$

the set of rankings $\pi$ that contain $\rho$ as a partial ranking. The set $\mathcal{P}(\rho)$ contains all rankings $\pi$ for which the items in $\mathcal{B}$ are ranked according to $\rho$, while the remaining items in $\mathcal{I} \setminus \mathcal{B}$ can be ranked arbitrarily.

Another special case of sub-rankings are top-k rankings, which only consider the items ranked in the first $k$ positions. For these, we introduce

$$\mathcal{T}_\mathcal{B}(\rho) = \{\pi \in \mathcal{S}_\mathcal{I} \mid \exists r \in \mathcal{S}_{\mathcal{I} \setminus \mathcal{B}} : \pi = \rho \succ r\}, \quad (8)$$

for each $\mathcal{B} \subseteq \mathcal{I}$ with $|\mathcal{B}| \geq 1$ and $\rho \in \mathcal{S}_\mathcal{B}$. In words, the set of rankings that coincide with $\rho$ on the top positions.

Here, the length of the top positions is determined by the cardinality of $\mathcal{B}$.

As an example to illustrate both concepts, consider $\mathcal{I} = \{1, 2, 3\}$ and $\mathcal{B} = \{1, 2\}$ with $\rho = 1 \succ 2$. Here, we obtain

$$\mathcal{P}_\mathcal{B}(\rho) = \{1 \succ 2 \succ 3, \ 1 \succ 3 \succ 2, \ 3 \succ 1 \succ 2\},$$
$$\mathcal{T}_\mathcal{B}(\rho) = \{1 \succ 2 \succ 3\}.$$

Notice that $\mathcal{T}(\rho) \subseteq \mathcal{P}(\rho)$ holds in general and $\mathcal{P}(\rho) = \mathcal{T}(\rho) = \{\rho\}$ only in the case of $\mathcal{B} = \mathcal{I}$. In the remainder of this work, we omit $\mathcal{B}$ from the subscripts of $\mathcal{P}(\rho)$ and $\mathcal{T}(\rho)$, where $\mathcal{B}$ is clear from context. As a matter of fact, the cardinality of $\mathcal{B}$ will play a more central role in what follows, and we assume throughout that $|\mathcal{B}| = k$. Accordingly, we refer to $\mathcal{P}$ as the *sub-k subsets* and $\mathcal{T}$ as the *top-k subsets*.

These two sets correspond to the respective modality, i.e., sub-k rankings or top-k rankings. The latter are of particular interest, as the evaluation of predictions in downstream tasks often considers top-rated items (top-k) or comparisons of specific subsets of items (sub-k), despite models in ProLR predicting full rankings. Alternative subsets, such as worst-k, are much less practically relevant and therefore not considered here. Next, we introduce a probability distribution $p \in \mathbb{P}(\mathcal{S}_\mathcal{I})$ suitable marginalizations for both subsets $\mathcal{P}$ and $\mathcal{T}$.

**Definition 3.** For a set of items $\mathcal{B} \subseteq \mathcal{I}$ with $|\mathcal{B}| = k$ and sub-ranking $\rho \in \mathcal{S}_\mathcal{B}$, the *sub-k marginal distribution* of a distribution $p \in \mathbb{P}(\mathcal{S}_\mathcal{I})$ is

$$p_{||\mathcal{P}}[\rho] = \sum\nolimits_{\pi \in \mathcal{P}(\rho)} p[\pi],$$

and the *top-k marginal distribution* is given by

$$p_{||\mathcal{T}}[\rho] = \sum\nolimits_{\pi \in \mathcal{T}(\rho)} p[\pi].$$

Note that the sub-k marginal distribution is an element of $\mathbb{P}(\mathcal{S}_\mathcal{B})$, i.e., a probability distribution on the set of rankings over the item set $\mathcal{B}$, while the top-k marginal distribution is an element of $\mathbb{P}(\cup_{\mathcal{B} \subset \mathcal{I}:|\mathcal{B}|=k} \mathcal{S}_\mathcal{B})$, i.e., a probability distribution on the set of all rankings of size $k$ consisting of elements in $\mathcal{I}$. We now consider calibration for the above marginal distributions with respect to a model $h$ that outputs distributions on $\mathcal{S}_\mathcal{I}$. First, we define the (full-rank) sub-k calibration again by conditioning on an entire probability distribution $q \in \mathbb{P}(\mathcal{S}_\mathcal{B})$. Here, we consider the sub-k marginal of $h(X)$ for all sub-k subsets.

**Definition 4.** A model $h \in \mathcal{H}$ is *sub-k* calibrated if for every $\mathcal{B} \subseteq \mathcal{I}$ and $|\mathcal{B}| = k$ it holds for all $\rho \in \mathcal{S}_\mathcal{B}$ that

$$\forall q \in \mathbb{P}(\mathcal{S}_\mathcal{B}) : \ P(\Pi_{||\mathcal{B}} = \rho \mid h_{||\mathcal{P}}(X) = q) = q[\rho], \quad (9)$$

where $\Pi_{||\mathcal{B}}$ denotes the projection of $\Pi$ onto the items in $\mathcal{B}$.

Note that for $k = 2$ we recover the modality of pairwise rankings, which is of special interest in Reinforcement

Learning from Human Feedback (RLHF). Similarly, we introduce *top-k* calibration by considering the top-k marginal of $h(X)$ for all top-k subsets.

**Definition 5.** A model $h \in \mathcal{H}$ is *top-k* calibrated if for every $\mathcal{B} \subseteq \mathcal{I}$ and $|\mathcal{B}| = k$ it holds for all $\rho \in \mathcal{S}_{\mathcal{B}}$ and all $q \in \mathbb{P}(\cup_{\mathcal{B} \subset \mathcal{I}: |\mathcal{B}| = k} \mathcal{S}_{\mathcal{B}})$ :

$$P(\Pi_{||\mathcal{B}} = \rho \mid h_{||\mathcal{T}}(X) = q) = q[\rho] \,. \quad (10)$$

In other words, the probability for all events, where the sub-k (top-k) marginalised observed ranking is $\rho$ conditional on the sub-k (top-k) marginalised model predicting $q$, is exactly the probability that $\rho$ has within $q$. For example, in cases where the model predicts the probability $p_{i,j}$ for the pairwise comparison $i \succ j$ (e.g., *rock* $\succ$ *pop*) and we consider these pairwise comparisons precisely, they occur in exactly $p_{i,j}$ of the cases. Another way to interpret sub-k or top-k calibration of a model $h$ is that its sub-k or top-k marginal distribution is full-rank calibrated on every subset $\mathcal{B} \subseteq \mathcal{I}$ of size $k$.

**Relationship of strong calibration notions in ProLR.** Intuitively, both sub-k and top-k calibration pose a weaker notion than full-rank calibration, as they both consider sub-rankings of size $k$ instead of full-rankings of size $m$. Indeed, one can show that full-rank calibration implies both sub-k and top-k calibration, reflected in the following theorem.

**Theorem 4.2.** *For every finite item set $\mathcal{I}$ and $k \leq m$:*

(i) *$h$ is full-rank calibrated $\Rightarrow$ $h$ is sub-k calibrated,*

(ii) *$h$ is full-rank calibrated $\Rightarrow$ $h$ is top-k calibrated.*

Considering the converse direction, we can show that neither sub-k nor top-k calibration implies full-rank calibration. Intuitively, this stems from the fact that both sub-k and top-k calibration consider only sub-rankings of size $k$, hence ignoring information about the remaining items. Therefore, modifications on the entire distribution $h(X) \in \mathbb{P}(\mathcal{S}_{\mathcal{I}})$ can cancel themselves out in the sub-k or top-k marginal distributions, yet lead to violations of full-rank calibration.

Table 1 shows an example of a model that is sub-$k$ calibrated yet not full-rank calibrated. An example of a top-$k$ calibrated model that is not full-rank calibrated is given in Table 3. More general results dealing with all values of $k$ are Theorems A.4 and A.5. Note that neither sub-k nor top-k calibration implies the other, as the former considers all rankings containing a specific sub-ranking, while the latter only considers rankings where the sub-ranking appears in the top-k positions. Again, modifications on the entire distribution $h(X) \in \mathbb{P}(\mathcal{S}_{\mathcal{I}})$ can cancel themselves out in one of the marginal distributions, yet lead to violations in the other. Thus, a model can be sub-k calibrated yet not top-k

*Table 1.* Counterexample showing that sub-2 calibration does not imply full-rank calibration. Sub-2 marginals satisfy calibration for both $x_1$ and $x_2$, whereas the full ranking probabilities violate calibration for both predictions.

| $\pi$ | $P(\Pi|x_1)$ | $P(\Pi|x_2)$ | $h(x_1)$ | $h(x_2)$ |
|---|---|---|---|---|
| $i_1 \succ i_2 \succ i_3$ | $1/6$ | $1/6$ | $2/6$ | $2/6$ |
| $i_1 \succ i_3 \succ i_2$ | $1/6$ | $1/6$ | $1/12$ | $1/12$ |
| $i_2 \succ i_1 \succ i_3$ | $1/6$ | $1/6$ | $1/12$ | $1/12$ |
| $i_2 \succ i_3 \succ i_1$ | $1/6$ | $1/6$ | $1/12$ | $1/12$ |
| $i_3 \succ i_1 \succ i_2$ | $1/6$ | $1/6$ | $1/12$ | $1/12$ |
| $i_3 \succ i_2 \succ i_1$ | $1/6$ | $1/6$ | $2/6$ | $2/6$ |
| **Sub-2 marginals** | | | | |
| $i_1 \succ i_2$ | $1/2$ | $1/2$ | $1/2$ | $1/2$ |
| $i_1 \succ i_3$ | $1/2$ | $1/2$ | $1/2$ | $1/2$ |
| $i_2 \succ i_3$ | $1/2$ | $1/2$ | $1/2$ | $1/2$ |

calibrated, and vice versa (see Theorems A.2 and A.3). This supports the view that both notions cover different aspects of calibration.

It remains to clarify how the weaker rankwise calibration relates to sub-k or top-k calibration. One can show that neither sub-k nor top-k calibration implies rankwise calibration, and vice versa. Thus, a model can be rankwise calibrated, yet not sub-k calibrated (Theorem A.7) or top-k calibrated (Theorem A.8). Similarly, a model can be sub-k calibrated but not rankwise calibrated (Theorem A.13) or can be top-k calibrated but not rankwise calibrated (Theorem A.14). These results can be intuitively explained: rankwise calibration conditions on the probability of a specific ranking, while sub-k and top-k calibration condition on marginal distributions over sub-rankings. Thus, roughly speaking, the notions are orthogonal and therefore do not imply anything about one another. An example of a model being rankwise calibrated but neither sub-k nor top-k calibrated is illustrated in Table 2.

**Weaker calibration notions in ProLR.** Similar to the transition from full-rank calibration to rankwise calibration, we can further weaken the conditions of sub-k and top-k calibration by conditioning only on the probability of a specific sub-ranking instead of the entire marginal distribution.

**Definition 6.** A model $h \in \mathcal{H}$ is *rankwise sub-k* calibrated if for every $\mathcal{B} \subseteq \mathcal{I}$ with $|\mathcal{B}| = k$ it holds for all $\pi \in \mathcal{S}_{\mathcal{B}}$ that

$$\forall \alpha \in [0,1] : \ P(\Pi_{||\mathcal{B}} = \pi \mid h_{||\mathcal{P}}(X)[\pi] = \alpha) = \alpha \,. \quad (11)$$

**Definition 7.** A model $h \in \mathcal{H}$ is *rankwise top-k* calibrated if for every $\mathcal{B}$ with $|\mathcal{B}| = k$ it holds for all $\pi \in \mathcal{S}_{\mathcal{B}}$ that

$$\forall \alpha \in [0,1] : \ P(\Pi_{||\mathcal{B}} = \pi \mid h_{||\mathcal{T}}(X)[\pi] = \alpha) = \alpha \,. \quad (12)$$

*Table 2.* Rankwise calibrated model that is neither sub-2 nor top-1 calibrated. Sub-2 marginals violate calibration for $x_2$ and top-1 marginals violate calibration for both $x_1$ and $x_2$.

| $\pi$ | $P(\Pi\|x_1)$ | $P(\Pi\|x_2)$ | $h(x_1)$ | $h(x_2)$ |
|---|---|---|---|---|
| $i_1 \succ i_2 \succ i_3$ | $1/3$ | $0$ | $1/3$ | $0$ |
| $i_1 \succ i_3 \succ i_2$ | $0$ | $1/3$ | $1/6$ | $1/6$ |
| $i_2 \succ i_1 \succ i_3$ | $1/3$ | $1/3$ | $1/3$ | $1/3$ |
| $i_2 \succ i_3 \succ i_1$ | $0$ | $1/3$ | $0$ | $1/3$ |
| $i_3 \succ i_1 \succ i_2$ | $1/3$ | $0$ | $1/6$ | $1/6$ |
| $i_3 \succ i_2 \succ i_1$ | $0$ | $0$ | $0$ | $0$ |
| **Sub-2 marginals** | | | | |
| $i_1 \succ i_2$ | $2/3$ | $1/3$ | $5/6$ | $1/3$ |
| $i_1 \succ i_3$ | $2/3$ | $2/3$ | $2/3$ | $1/2$ |
| $i_2 \succ i_3$ | $2/3$ | $2/3$ | $2/3$ | $2/3$ |
| **Top-1 marginals** | | | | |
| $i_1 \succ \{i_2, i_3\}$ | $1/3$ | $1/3$ | $1/2$ | $1/6$ |
| $i_2 \succ \{i_1, i_3\}$ | $1/3$ | $2/3$ | $1/3$ | $1/3$ |
| $i_3 \succ \{i_1, i_2\}$ | $1/3$ | $0$ | $1/6$ | $1/6$ |

Obviously, it holds that sub-k calibration implies rankwise sub-k calibration, and similarly for top-k and rankwise top-k calibration, as formulated in the following theorem.

**Theorem 4.3.** *For every finite item set $\mathcal{I}$ and $k \leq m$:*

*(i) $h$ is sub-k calibrated $\Rightarrow$ $h$ is rankwise sub-k calibrated*

*(ii) $h$ is top-k calibrated $\Rightarrow$ $h$ is rankwise top-k calibrated*

Both rankwise sub-k and rankwise top-k calibration pose a weaker notion than their full marginal counterparts, as they consider only conditioning on the probability of a specific sub-ranking instead of the entire marginal distribution. Due to this, one can show that neither rankwise sub-k nor rankwise top-k calibration implies (full-rank) sub-k or top-k calibration, respectively. Interestingly, in contrast to the stronger notion of full-rank calibration, rankwise calibration implies neither sub-$k$ calibration nor top-$k$ calibration. Table 1 shows a sub-2 calibrated model that is not rankwise calibrated, while Table 3 shows a top-1 calibrated model that is not rankwise calibrated.

Moreover, one can show that neither rankwise sub-k nor rankwise top-k calibration implies rankwise calibration, which again holds for any model class. The intuition behind this is similar to the case of full-rank calibration not being implied by sub-k or top-k calibration, as rankwise sub-k and rankwise top-k calibration consider only sub-rankings of size $k$. Further, the conditioning is only on the probability of a specific sub-ranking, hence modifications on the entire distribution $h(X) \in \mathbb{P}(\mathcal{S}_\mathcal{I})$ can cancel themselves out in the conditioning of a specific sub-ranking, yet remain in the rankwise conditioning. Formally, this is shown in

*Table 3.* Top-1 calibrated model that is not rankwise calibrated. Top-1 marginals are identical for both $x_1$ and $x_2$, whereas the model violates rankwise calibration for both predictions. Note that the model is also not full-rank calibrated.

| $\pi$ | $P(\Pi\|x_1)$ | $P(\Pi\|x_2)$ | $h(x_1)$ | $h(x_2)$ |
|---|---|---|---|---|
| $i_1 \succ i_2 \succ i_3$ | $1/6$ | $1/6$ | $2/6$ | $2/6$ |
| $i_1 \succ i_3 \succ i_2$ | $1/6$ | $1/6$ | $0$ | $0$ |
| $i_2 \succ i_1 \succ i_3$ | $1/6$ | $1/6$ | $2/6$ | $2/6$ |
| $i_2 \succ i_3 \succ i_1$ | $1/6$ | $1/6$ | $0$ | $0$ |
| $i_3 \succ i_1 \succ i_2$ | $1/6$ | $1/6$ | $2/6$ | $2/6$ |
| $i_3 \succ i_2 \succ i_1$ | $1/6$ | $1/6$ | $0$ | $0$ |
| **Top-1 marginals** | | | | |
| $i_1 \succ i_2$ | $1/3$ | $1/3$ | $1/3$ | $1/3$ |
| $i_1 \succ i_3$ | $1/3$ | $1/3$ | $1/3$ | $1/3$ |
| $i_2 \succ i_3$ | $1/3$ | $1/3$ | $1/3$ | $1/3$ |

Theorems A.11 and A.12 in Appendix A. Notice that one can show that neither rankwise sub-k nor rankwise top-k calibration implies the other, as shown in Theorems A.9 and A.10. The overall relationships between the different notions of calibration are shown in Figure 1.

## 5. Experiments

Our experiments are divided into two parts. First, we investigate the calibration with respect to our definitions of existing learners for ProLR. In the second part, we investigate the calibration of reward models in a popular benchmark for RLHF. In both parts, we consider the same evaluation metrics, which we explain next prior to presenting the experimental setup and results.

**Evaluation Metrics.** We use the expected calibration error (ECE) (Silva Filho et al., 2023) to evaluate the different notions of calibration. Simply put, we bin the predicted probabilities into $B$ equally sized bins, and for each bin we compute the absolute difference between the average predicted probability and the empirical frequency of the ranking. Here, a ranking could either be the ranking over all $m$, or only over a subset of $k$ items. To compute the empirical frequency of the event in each bin, we consider all instances $x_i$ for which the predicted probability of the event falls into the respective bin. Given a binning $bin_1, \ldots, bin_B$ of $[0, 1]$, a ranking $\tau$ and data $\mathcal{D} = \{(x_i, \pi_i)\}_{i=1}^n$, we compute the $ECE_\tau$ for rankwise calibration as

$$\text{ECE}_\tau = \sum_{b=1}^{B} \frac{|I_b|}{n} |\text{freq}(I_b) - \text{conf}(I_b)|,$$

where $I_b = \{i \in \{1, \ldots, n\} \mid h(x_i)[\tau] \in \text{bin}_b\}$ is the set of instances in bin $b$, $\text{freq}(I_b) = \frac{1}{|I_b|} \sum_{i \in I_b} \mathbb{1}\{\pi_i = \tau\}$ the accuracy in bin $b$, and $\text{conf}(I_b) = \frac{1}{|I_b|} \sum_{i \in I_b} h(x_i)[\tau]$ the

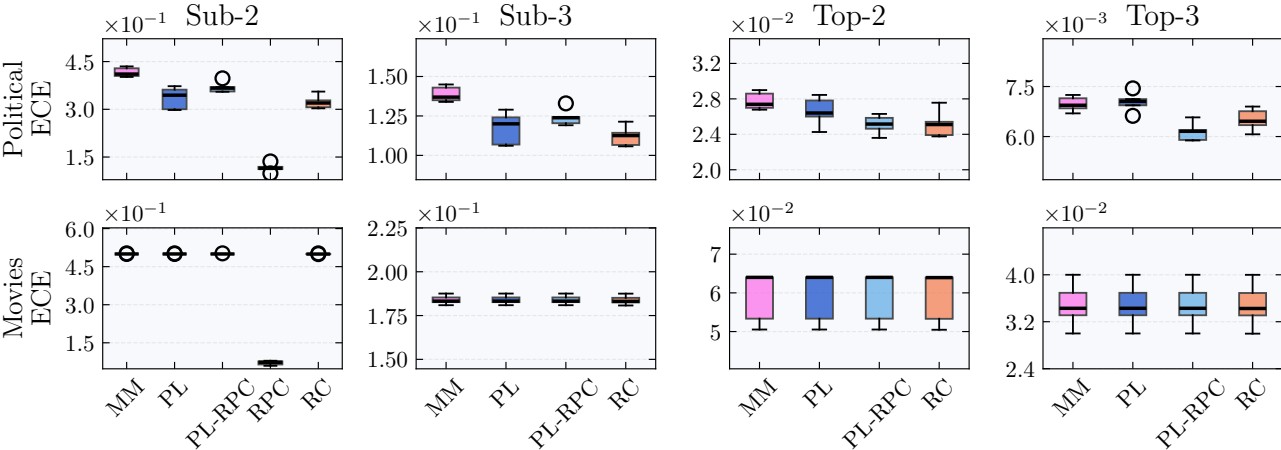

*Figure 2.* Rankwise sub-2,sub-3,top-2, and top-3 ECE on *political* and *movies*, considering only the 95% most occurring rankings.

average predicted probability in bin $b$. Similarly, for rankwise sub-k and top-k calibration, we compute the $\text{ECE}_\rho$ for a sub-ranking $\rho$ of size $k$ and average over all sub-rankings $\mathcal{P}$, $\mathcal{T}$. We refer to Appendix E.1 for further details and critical discussion.

### 5.1. Calibration of Label Ranking Models

**Methods.** We investigate the different notions of calibration for popular label ranking models such as Plackett–Luce (PL) (Cheng et al., 2010), Mallows Model (MM) (Cheng & Hüllermeier, 2009) and Ranking by Pairwise Comparison (RPC) (Hüllermeier & Furnkranz, 2004). Additionally, we consider a neural network classifier in which each full ranking corresponds to a single class, which we denote as RankClassifier (RC). Finally, we consider the PL-RPC model, obtaining Plackett–Luce parameters from an RPC model (see Section 3), as RPC calibration can only be investigated for sub-2 (see Appendix C). Details on the models, their implementation, and source code are given in Appendix C.

**Datasets.** We consider popular semi-synthetic and real-world benchmark datasets from the label ranking literature (Cheng et al., 2009). The following experiments are focused on the latter (*movies* and *political*), while Section E.2 reports results for the semi-synthetic datasets. More details on all datasets are provided in Appendix B.

**Setup and Results.** We evaluate Expected Calibration Error (ECE) on the *movies* and *political* label ranking datasets using 5-fold cross-validation to obtain error bounds. Here, we consider only rankings of length $k = 3$. We refer to Section E.2 for the complete results. As shown in Figure 2, RPC achieves the strongest calibration performance for pairwise rankings, which can be attributed to its pairwise learning paradigm. The effect of rank breaking (Soufiani et al., 2013)

is also evident: although RPC itself is well calibrated for $k = 2$, the Plackett–Luce–RPC variant shows poor calibration. Both Plackett–Luce and Mallows are generally poorly calibrated across the considered settings. RankClassifier displays mixed behavior, being well calibrated in some cases while showing substantial miscalibration in others, such as sub-2 rankings on the *political* and *movies* datasets (see Figure 2). A similar pattern is observed on the semi-synthetic datasets reported in Appendix E.2, underscoring the need for calibration techniques tailored to probabilistic label ranking. As the number of ranking classes grows factorially with $k$, the ECE becomes increasingly uninformative. We refer to Section 6 and Appendix E.1 for further details.

### 5.2. Calibration of RLHF Reward Models

Reward models are central to RLHF, guiding policy optimization by scoring response quality. Such models are often trained with Bradley–Terry (Ouyang et al., 2022; Kaufmann et al., 2025b) and evaluated on benchmarks such as Reward-Bench2 (Malik et al., 2025), which is structured as a top-1 task.[2] We evaluate reward models from RewardBench2 in our experiments. Each reward model predicts utilities for four candidate responses, yielding top-1 probabilities via the PL model (see Appendix D for details).

Figure 3 shows the ECE of the ten most calibrated models, shown in Figure 3a, and the correlation between ECE and the individual RewardBench2 categories, shown in Figure 3b. For the former, we observe that well-calibrated models also tend to achieve high accuracy. Regarding the correlations, the strongest association is observed for the leaderboard score; however, this correlation remains imperfect, leaving room for models that perform well according to one metric but not the other. We also observe notable

---

[2]The "Ties" category does not fit this framing and is excluded, see Appendix D.

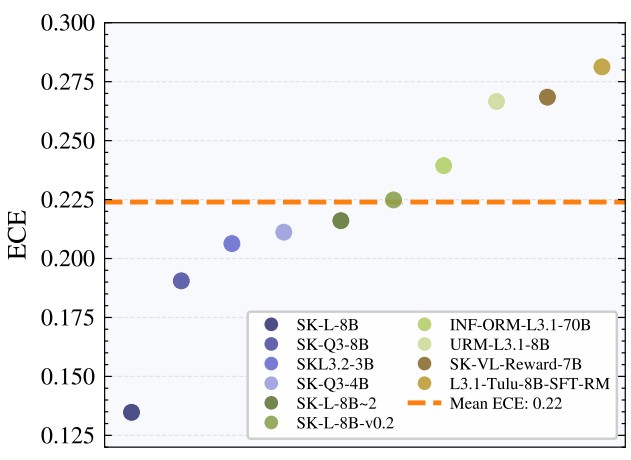

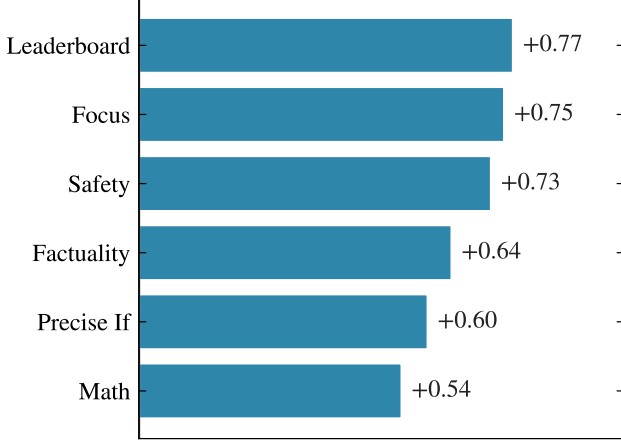

*(a)* Top-1 ECE of the top-10 most calibrated RewardBench2 models. We abbreviate Llama with "L" and Skywork with "SK".

*(b)* Correlation of rankings obtained by top-1 ECE and categories of RewardBench2.

*Figure 3.* Top-1 ECE results and their correlation with RewardBench2 categories.

differences across categories. Focus[3] and Safety exhibit the strongest correlations, whereas Math and Precise If[4] exhibit the weakest. One possible explanation is that Focus and Safety allow for a more gradual notion of correctness, enabling model confidence to better track response quality. In contrast, Math and Precise If are closer to step-function tasks, where small errors can lead to complete failure, making confidence less predictive of correctness.

## 6. Discussion, Conclusion and Future Work

**Practical Considerations.** As the number of classes increases, the exact computation of the ECE becomes infeasible. Moreover, it is difficult to envisage a scenario in which a practitioner would be concerned with whether the predicted probability of an extremely improbable ranking of, say, ten items is 0.00001 or 0.00002. For our practical investigation, we therefore restrict the computation to those rankings that together account for up to $95\%$ of the ranking occurrences in the datasets. For downstream tasks, one should either focus on the rankings of greatest interest or use weaker notions of calibration, such as sub-k and top-k calibration. If calibration on all rankings is nevertheless required, we refer to Appendix E.1, which discusses (i) alternative metrics for such cases and (ii) a bottom-to-top approach: first investigating calibration for weaker, more tractable notions and then considering more complex ones.

**Conclusion.** We introduced a hierarchy of calibration notions for probabilistic label ranking, from full-ranking to pairwise and top-k calibration. We established theoretical relationships between the different notions and empirically studied the (non-)calibration of popular label ranking models on real-world datasets as well as reward models in RLHF.

**Future Work.** Given the hierarchy presented in Figure 1, a natural next step is to develop tailored calibration methods and metrics for label ranking. Furthermore, the impact of calibration on downstream tasks, such as fine-tuning large language models, should be investigated.

## Acknowledgements

Timo Kaufmann and Eyke Hüllermeier gratefully acknowledge funding by the German Research Foundation (Deutsche Forschungsgemeinschaft, DFG), project number 467367360. The authors acknowledge support and computational resources from the Munich Center of Machine Learning (MCML). For the CPU-based synthetic experiments, the authors gratefully acknowledge the computational and data resources provided by the Leibniz Supercomputing Centre (www.lrz.de).

## Impact Statement

The most important results are theoretical in nature and relate to foundational research. These may lead to follow-up work with potential societal implications, particularly in the area of reinforcement learning from human feedback and its close connection to the fine-tuning of LLMs. Although there are many potential societal consequences of our work, there is none which we feel must be specifically highlighted here.

---

[3]Focus tests whether the RM can detect responses that are off-topic or fail to address the question asked.

[4]Precise If requires the model to follow specific verifiable constraints in its response.

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

# Appendix

# A. Proofs

This section is divided into two parts: The first part deals with the theorems formulated in the main paper and shows implications, while the second part deals with results showing the exclusiveness of the notions with respect to one another.

## A.1. Implication Results

**Theorem 4.1** For an item set $\mathcal{I}$ and rankings $\mathcal{S}_\mathcal{I}$, it holds

$$h \text{ is full-rank calibrated } \Rightarrow h \text{ is rankwise calibrated}$$

*Proof of Theorem 4.1.* If $h$ is full-rank calibrated, then by Definition 1 it holds for all $\pi \in \mathcal{S}_\mathcal{I}$:

$$\forall p \in \mathbb{P}(\mathcal{S}_\mathcal{I}) : \ P(\Pi = \pi \mid h(X) = p) = p[\pi] \,.$$

Now let $\pi \in \mathcal{S}_\mathcal{I}$ and $\alpha \in [0,1]$ be fixed but arbitrary. Then,

$$P(\Pi = \pi \mid h(X)[\pi] = \alpha) = \int_{\{p:\, p[\pi]=\alpha\}} P(\Pi = \pi \mid h(X) = p) \, dP(h(X) = p \mid h(X)[\pi] = \alpha) \tag{13}$$

$$= \int_{\{p:\, p[\pi]=\alpha\}} p[\pi] \, dP(h(X) = p \mid h(X)[\pi] = \alpha) \tag{14}$$

$$= \int_{\{p:\, p[\pi]=\alpha\}} \alpha \, dP(h(X) = p \mid h(X)[\pi] = \alpha) \tag{15}$$

$$= \alpha \,. \tag{16}$$

Step (13) is the law of total probability, decomposing over the conditional distribution of $h(X)$ given $h(X)[\pi] = \alpha$. Step (14) applies full-rank calibration. Since $p[\pi] = \alpha$ throughout the domain of integration, the integrand is constant, and the conditional measure integrates to one, yielding (16). □

**Theorem 4.2** For every finite item set $\mathcal{I}$ and $k \leq m$:

(i) $h$ is full-rank calibrated $\Rightarrow h$ is sub-k calibrated

(ii) $h$ is full-rank calibrated $\Rightarrow h$ is top-k calibrated

*Proof of Theorem 4.2 (i).* Let $k \leq m$, $\mathcal{B} \subseteq \mathcal{I}$ with $|\mathcal{B}| = k$. Additionally, let $q \in \mathbb{P}(\mathcal{S}_\mathcal{B})$ and $\rho \in \mathcal{S}_\mathcal{B}$ arbitrary but fixed. We are given $h : \mathcal{X} \to \mathbb{P}(\mathcal{S}_\mathcal{I})$, which we assume to be full-rank calibrated as in Definition 1. Our goal is to show that it holds:

$$P(\Pi_{||\mathcal{B}} = \rho \mid h_{||\mathcal{P}} = q) = q[\rho] \,.$$

Let $\tau_1, \ldots, \tau_c \in \mathcal{S}_\mathcal{I}$, where $c = \frac{m!}{k!}$, be all rankings for which $\tau_i \in \mathcal{P}(\rho)$. Then it holds:

$P(\Pi_{||\mathcal{B}} = \rho \mid h_{||\mathcal{P}}(X) = q)$

$$= \int_{p:p_{||\mathcal{P}}=q} P(\Pi_{||\mathcal{B}} = \rho \mid h(X) = p, h_{||\mathcal{P}}(X) = p_{||\mathcal{P}}) dP(h(X) = p \mid h_{||\mathcal{P}}(X) = q) \quad (17)$$

$$= \int_{p:p_{||\mathcal{P}}=q} P(\Pi_{||\mathcal{B}} = \rho \mid h(X) = p) dP(h(X) = p \mid h_{||\mathcal{P}}(X) = q) \quad (18)$$

$$= \int_{p:p_{||\mathcal{P}}=q} \sum_{i=1}^{c} P(\Pi = \tau_i \mid h(X) = p) dP(h(X) = p \mid h_{||\mathcal{P}}(X) = q) \quad (19)$$

$$= \int_{p:p_{||\mathcal{P}}=q} \sum_{i=1}^{c} p[\tau_i] dP(h(X) = p \mid h_{||\mathcal{P}}(X) = q) \quad (20)$$

$$= \int_{p:p_{||\mathcal{P}}=q} q[\rho] dP(h(X) = p \mid h_{||\mathcal{P}}(X) = q) \quad (21)$$

$$= q[\rho] \int_{p:p_{||\mathcal{P}}=q} 1 dP(h(X) = p \mid h_{||\mathcal{P}}(X) = q) \quad (22)$$

$$= q[\rho] \quad (23)$$

In (17) and (18) we use the law of total probability. Since $h_{||\mathcal{P}}$ is a deterministic function of $h$, conditioning on both $h(X) = p$ and $h_{||\mathcal{P}}(X) = q$ is equivalent to conditioning on $h(X) = p$ when $p_{||\mathcal{P}} = q$. In (19), we use that $\{\Pi_\mathcal{B} = \rho\}$ is the disjoint union of the events $\{\Pi = \tau_i\}$. Step (20) uses the definition of full-rank calibration (Definition 1). Step (21) uses Definition 3 as well as the condition on the domain of the integral, while Step (22) is by linearity of the integral and $q[\rho]$ being a constant. Finally, the conditional distribution $P(h(X) = p \mid h_{||\mathcal{P}}(X) = q)$ integrates to 1 in (23). $\qquad\square$

*Proof of Theorem 4.2 (ii).* The proof is analogous to Theorem 4.2 (i) by replacing $\mathcal{P}$ by $\mathcal{T}$, taking $c = (m-k)!$ and using $q \in \mathbb{P}(\cup_{\mathcal{B} \subset \mathcal{I}:|\mathcal{B}|=k} \mathcal{S}_\mathcal{B})$. $\qquad\square$

**Theorem 4.3** For every finite item set $\mathcal{I}$ and $k \leq m$:

(i) $h$ is sub-k calibrated $\Rightarrow h$ is rankwise sub-k calibrated

(ii) $h$ is top-k calibrated $\Rightarrow h$ is rankwise top-k calibrated

*Proof of Theorem 4.3 (i).* Let $h$ be sub-k calibrated for $k \leq m$ arbitrary but fixed. Then by Definition 4 for every $\mathcal{B} \subseteq \mathcal{I}$ and $|\mathcal{B}| = k$ it holds for all $\tau \in \mathcal{S}_\mathcal{B}$:

$$\forall q \in \mathbb{P}(\mathcal{S}_\mathcal{B}): \ P(\Pi_{||\mathcal{B}} = \tau \mid h_{||\mathcal{P}}(X) = q) = q[\tau].$$

Now let $\tau \in \mathcal{S}_\mathcal{B}$ and $\alpha \in [0, 1]$ be fixed but arbitrary. Then,

$$P(\Pi_{||\mathcal{B}} = \tau \mid h_{||\mathcal{P}}(X)[\tau] = \alpha) = \int_{\{q:\, q[\tau]=\alpha\}} P(\Pi_{||\mathcal{B}} = \tau \mid h_{||\mathcal{P}}(X) = q) \, dP(h_{||\mathcal{P}}(X) = q \mid h_{||\mathcal{P}}(X)[\tau] = \alpha) \quad (24)$$

$$= \int_{\{q:\, q[\tau]=\alpha\}} q[\tau] \, dP(h_{||\mathcal{P}}(X) = q \mid h_{||\mathcal{P}}(X)[\tau] = \alpha) \quad (25)$$

$$= \int_{\{q:\, q[\tau]=\alpha\}} \alpha \, dP(h_{||\mathcal{P}}(X) = q \mid h_{||\mathcal{P}}(X)[\tau] = \alpha) \quad (26)$$

$$= \alpha. \quad (27)$$

Step (24) is the law of total probability, decomposing over the conditional distribution of $h_{||\mathcal{P}}(X)$ given $h_{||\mathcal{P}}(X)[\tau] = \alpha$. Step (25) applies sub-k calibration. Since $q[\tau] = \alpha$ throughout the domain of integration, the integrand is constant (26), and the conditional measure integrates to one, yielding (27). $\qquad\square$

*Proof of Theorem 4.3 (ii).* The proof is analogous to Theorem 4.3 (i), replacing $\mathcal{P}$ by $\mathcal{T}$ and considering the law of total probability over $q \in \mathbb{P}(\cup_{\mathcal{B} \subset \mathcal{I}:|\mathcal{B}|=k} \mathcal{S}_\mathcal{B})$. $\qquad\square$

## A.2. Exclusiveness Results

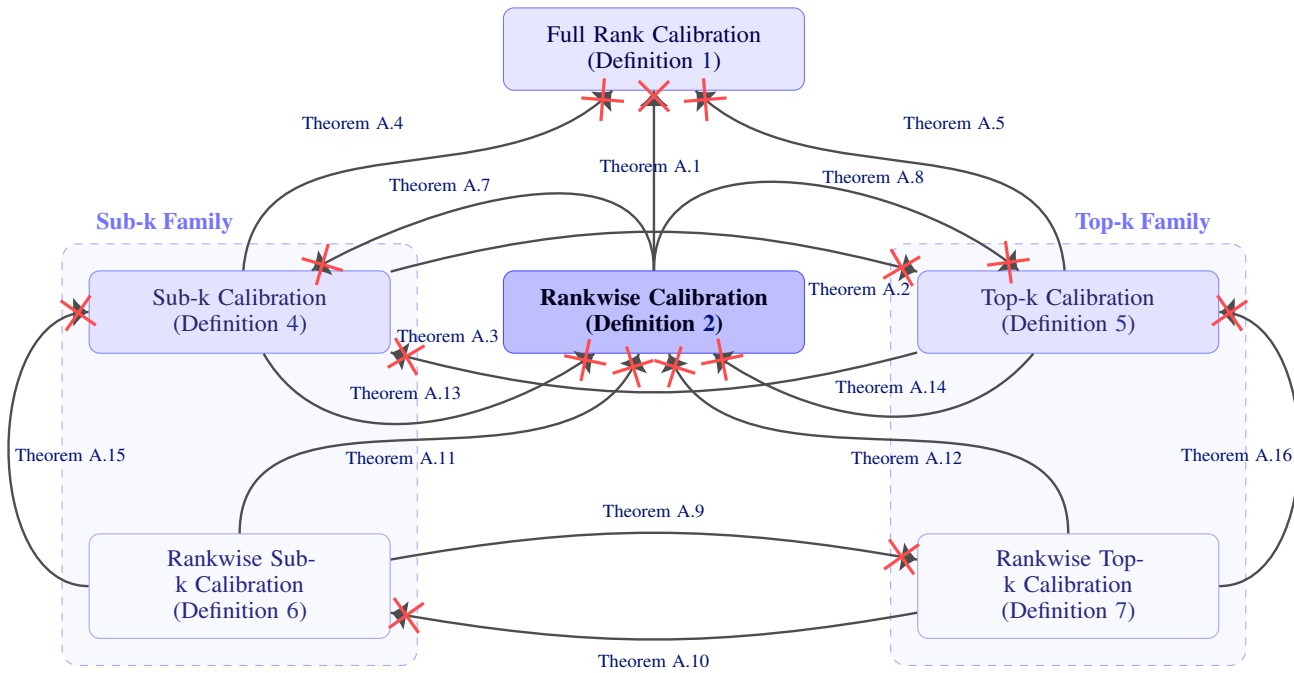

*Figure 4.* Overview of the calibration definitions and their exclusiveness relationships.

**Theorem A.1.** *Let $\mathcal{X}$ consist of at least two elements. If there exist two data points $x_i, x_j \in \mathcal{X}$ with $i \neq j$ such that for two rankings $\pi_1, \pi_2 \in \mathcal{S}_{\mathcal{I}}$ it holds that*

$$h(x_i)[\pi_1] = h(x_j)[\pi_1] \wedge h(x_i)[\pi_2] = h(x_j)[\pi_2]$$

*and $P(X = x_i \mid h(x_i)[\alpha] = \alpha) = P(X = x_j \mid h(x_j)[\alpha] = \alpha)$ for some $\alpha \in [0, 1]$, then it holds:*

$$\exists h \in \mathcal{X} \to \mathbb{P}(\mathcal{S}_{\mathcal{I}}) : h \text{ rankwise calibrated } \wedge h \text{ not } full\text{-}rank \text{ calibrated}.$$

*Proof of Theorem A.1.* Without loss of generality, let $\mathcal{X}$ consist of $2 \leq n < \infty$ data points, and let $h \in \mathcal{H}$ be full-rank calibrated, thus also rankwise calibrated by Theorem 4.1. The proof can be easily extended to the case of uncountable infinite sets $\mathcal{X}$ (see the proof of Theorem A.2). Let $\alpha, \alpha' \in [0, 1]$, $p \in \mathbb{P}(\mathcal{S}_{\mathcal{I}})$ be arbitrary but fixed and $\pi, \tau \in \mathcal{S}_{\mathcal{I}}$ with $\pi \neq \tau$. We choose the data points in $\mathcal{X}$ such that there exist two $x_i, x_j \in \mathcal{X}$ for which $x_i, x_j \in \mathcal{G} = \{x \in \mathcal{X} \mid h(x)[\pi] = \alpha \wedge h(x)[\tau] = \alpha'\}$, $x_i \in \mathcal{F} = \{x \in \mathcal{X} \mid h(x) = p\}$, $x_j \notin \mathcal{F}$ and $P(X = x_i \mid h(x_i)[\pi] = \alpha) = P(X = x_j \mid h(x_j)[\pi] = \alpha)$. Intuitively, $\mathcal{G}$ is the set of data points for which $h$ assigns the probability $\alpha$ to the ranking $\pi$, and $\mathcal{F}$ is the set of points for which the marginals $h_{\|\mathcal{P}}$ are identical to the probability vector $q$. Importantly, we have chosen $x_i, x_j$ in such a way that they have identical probability for $\pi, \tau$ but differ in at least one other ranking. As a next step we construct new data points $z_1, \ldots, z_n$ based on $x_1, \ldots, x_n$, for which $h$ is rankwise calibrated yet not full-rank calibrated. For $z_i, z_j$ we construct

(i)          $P(\Pi = \pi \mid X = z_i) = P(\Pi = \pi \mid X = x_i) + \delta$    $P(\Pi = \pi \mid X = z_j) = P(\Pi = \pi \mid X = x_j) - \delta$

(ii)         $P(\Pi = \tau \mid X = z_i) = P(\Pi = \tau \mid X = x_i) - \delta$    $P(\Pi = \tau \mid X = z_j) = P(\Pi = \tau \mid X = x_j) + \delta$

(iii) $\forall r \in \mathcal{S}_{\mathcal{I}} \setminus \{\tau, \pi\}:$    $P(\Pi = r \mid X = z_i) = P(\Pi = r \mid X = x_i)$       $P(\Pi = r \mid X = z_j) = P(\Pi = r \mid X = x_j)$

We choose $\delta$ in such a way that the above adjustments again lead to valid probability distributions, i.e.

$$\max\left(P(\Pi = \tau \mid X = x_i), P(\Pi = \pi \mid X = x_j)\right) \leq \delta \leq 1 - \max\left(P(\Pi = \pi \mid X = x_i), P(\Pi = \tau \mid X = x_j)\right).$$

For all other data points $l \in \{1, \ldots, n\} \setminus \{i, j\}$ we have $P(\Pi \mid X = z_l) = P(\Pi \mid X = x_l)$. Obviously, $h$ is still rankwise calibrated on $z_1, \ldots, z_n$ as in group $\mathcal{G}$ the changes $\delta$ and $-\delta$ cancel themselves out. Indeed, by writing $P_{z_1, \ldots, z_n}$ and $P_{x_1, \ldots, x_n}$ to indicate the considered feature space, we obtain

$$P_{z_1, \ldots, z_n}(\Pi = \pi \mid h(X)[\pi] = \alpha) = \sum_{\substack{z \in \mathcal{Z} \\ h(z)[\pi] = \alpha}} P(\Pi = \pi \mid X = z) P(X = z \mid h(X)[\pi] = \alpha) \tag{28}$$

$$= \sum_{\substack{z \in \mathcal{Z} \setminus \{z_i, z_j\} \\ h(z)[\pi] = \alpha}} P(\Pi = \pi \mid X = z) P(X = z \mid h(X)[\pi] = \alpha)$$

$$+ P(\Pi = \pi \mid X = z_i) P(X = z_i \mid h(X)[\pi] = \alpha)$$
$$+ P(\Pi = \pi \mid X = z_j) P(X = z_j \mid h(X)[\pi] = \alpha) \tag{29}$$

$$= \sum_{\substack{x \in \mathcal{X} \setminus \{x_i, x_j\} \\ h(x)[\pi] = \alpha}} P(\Pi = \pi \mid X = x) P(X = x \mid h(X)[\pi] = \alpha)$$

$$+ \big( P(\Pi = \pi \mid X = x_i) + \delta \big) P(X = x_i \mid h(X)[\pi] = \alpha)$$
$$+ \big( P(\Pi = \pi \mid X = x_j) - \delta \big) P(X = x_j \mid h(X)[\pi] = \alpha) \tag{30}$$

$$= \sum_{\substack{x \in \mathcal{X} \\ h(x)[\pi] = \alpha}} P(\Pi = \pi \mid X = x) P(X = x \mid h(X)[\pi] = \alpha) \tag{31}$$

$$= P_{x_1, \dots, x_n}(\Pi = \pi \mid h(X)[\pi] = \alpha) \tag{32}$$

$$= \alpha \tag{33}$$

Here, Steps (28) and (32) are by the law of total probability. Steps (29) and (30) use straightforward rewrites, while (31) uses the assumption on $x_i$ and $x_j$. Finally, using that $h$ is rankwise calibrated on $\mathcal{X}$ yields (31).

More interestingly, $h$ is not full-rank calibrated, as the $\delta$ changes do not cancel themselves. Proceeding similarly as in the display before, one obtains

$$P_{z_1, \dots, z_n}(\Pi = \pi \mid h(X) = q) = \sum_{\substack{x \in \mathcal{X} \setminus \{x_i, x_j\} \\ h(X) = q}} P(\Pi = \pi \mid X = x) P(X = x \mid h(X) = q)$$

$$+ \big( P(\Pi = \pi \mid X = x_i) + \delta \big) P(X = x_i \mid h(X) = q) \tag{34}$$

$$= \sum_{\substack{x \in \mathcal{X} \\ h(X) = q}} P(\Pi = \pi \mid X = x) P(X = x \mid h(X) = q)$$

$$+ \delta P(X = x_i \mid h(X) = q) \tag{35}$$

$$= P_{x_1, \dots, x_n}(\Pi = \pi \mid h(X) = q) + \delta P(X = x_i \mid h(X) = q) \tag{36}$$

$$= q[\pi] + \delta P(X = x_i \mid h(X) = q) \tag{37}$$

$$\neq q[\pi] \tag{38}$$

Here, we use in (37) that $h$ is full-rank calibrated on $\mathcal{X}$ and that $x_i \in \mathcal{F}$ for (38). $\qquad \square$

**Theorem A.2.** *Assume that the number of items $m \geq 3$. Then it holds:*

$$\forall k < m : \ \exists h \in \mathcal{X} \to \mathbb{P}(\mathcal{S}_{\mathcal{I}}) : h \ \text{sub-k calibrated} \ \wedge h \ \text{not top-k calibrated.}$$

*Proof of Theorem A.2.* Let $\mathcal{B} \subseteq \mathcal{I}$ be a set of items, $\rho \in \mathcal{S}_{\mathcal{B}}$, and let $p \in \mathbb{P}(\mathcal{S}_{\mathcal{I}})$ and $q \in \mathbb{P}(\mathcal{S}_{\mathcal{B}})$ be arbitrary but fixed. Without loss of generality [5] let $\tau_1 = i_1 \succ i_2 \succ \cdots \succ i_m$ and $\tau_2 = i_m \succ \cdots \succ i_2 \succ i_1$. Let $h$ be full-rank calibrated on $\mathcal{X}$ and such that there is a subset $\mathcal{X}_0 = \{x_1, \dots, x_n\}$ of $\mathcal{X}$ with $1 < n < \infty$ points such that $h(x)$ is non-degenerated for each $x \in \mathcal{X}_0$. Recall that sub-k calibration (9) satisfies

$$P(\Pi_{||\mathcal{B}} = \tau \mid h_{||\mathcal{P}}(X) = q) = q[\tau] = \sum_{\substack{x \in \mathcal{X} \\ h_{||\mathcal{P}}(x) = q}} P(\Pi_{||\mathcal{B}} = \tau \mid X = x) \, P(X = x \mid h_{||\mathcal{P}}(X) = q)$$

and top-k calibration (10) satisfies

$$P(\Pi_{||\mathcal{B}} = \rho \mid h_{||\mathcal{T}}(X) = q) = q[\rho] = \sum_{\substack{x \in \mathcal{X} \\ h_{||\mathcal{T}}(x) = q}} P(\Pi_{||\mathcal{B}} = \rho \mid X = x) \, P(X = x \mid h_{||\mathcal{T}}(X) = q).$$

---

[5] Choosing different $\tau_1, \tau_2$ is possible by relabeling the items, such that $\tau_1 = i_1 \succ i_2 \succ \dots$ and $\tau_2 = i_m \succ i_{m-1} \succ \dots$.

Particularly as $k < m$ it holds $\frac{m!}{k!} = |\mathcal{P}(\rho)| > |\mathcal{T}(\rho)| = (m-k)!$. We now construct a new feature subset $\mathcal{Z} = \{z_1, \ldots, z_n\}$ such that

$$P(X = z_i \mid h_{||\mathcal{T}}(X) = q) = P(X = x_i \mid h_{||\mathcal{T}}(X) = q) > 0 \tag{39}$$

and

$$P(\Pi = \pi \mid X = z_i) = \begin{cases} P(\Pi = \pi \mid X = x_i) + \delta & \text{if } \pi = \tau_1 \text{ or } \pi = \tau_2 \\ P(\Pi = \pi \mid X = x_i) - \frac{\delta}{|\mathcal{P}(\rho)|-1} & \text{if } \pi \in \mathcal{P}(\rho) \setminus \{\tau_1, \tau_2\} \\ P(\Pi = \pi \mid X = x_i) - \frac{\delta}{m!-|\mathcal{P}(\rho)|-1} & \text{otherwise} \end{cases} \tag{40}$$

for all $1 \leq i \leq n$. Here $\delta$ is chosen such that

$$0 < \delta \leq \min\Big( \min_{\substack{i \leq n \\ \pi \in \{\tau_1, \tau_2\}}} 1 - P(\Pi = \pi \mid X = x_i), \min_{\substack{\pi \in \mathcal{P}(\rho) \\ \tau \notin \{\tau_1, \tau_2\}}} (|\mathcal{P}(\rho)| - 1)P(\Pi = \pi \mid X = x_i)$$

$$\min_{\substack{\pi \in \mathcal{S}_\mathcal{I} \setminus \mathcal{P}(\rho) \\ \tau \notin \{\tau_1, \tau_2\}}} (m! - |\mathcal{P}(\rho)| - 1)P(\Pi = \pi \mid X = x_i)\Big).$$

This ensures that the probability on the left hand side of (40) takes only values between 0 and 1. The redistribution of the probability mass is designed in such a way that the left hand side of (40) is still a valid distribution summing to 1. Indeed, note that either $\tau_1$ or $\tau_2$ can be an element of $\mathcal{P}(\rho)$. Indeed, for each $z_i \in \mathcal{Z}$ we have

$$\sum_{\pi \in \mathcal{S}_\mathcal{I}} P(\Pi = \pi \mid X = z_i)$$

$$= P(\Pi = \tau_1 \mid X = x_i) + P(\Pi = \tau_2 \mid X = x_i) + 2\delta + \sum_{\substack{\pi \in \mathcal{P}(\rho) \\ \tau \notin \{\tau_1, \tau_2\}}} P(\Pi = \pi \mid X = z_i) + \sum_{\substack{\pi \in \mathcal{S}_\mathcal{I} \setminus \mathcal{P}(\rho) \\ \tau \notin \{\tau_1, \tau_2\}}} P(\Pi = \pi \mid X = z_i)$$

$$= \sum_{\pi \in \mathcal{S}_\mathcal{I}} P(\Pi = \pi \mid X = x_i) + 2\delta - \sum_{\substack{\pi \in \mathcal{P}(\rho) \\ \tau \notin \{\tau_1, \tau_2\}}} \frac{\delta}{|\mathcal{P}(\rho)| - 1} - \sum_{\substack{\pi \in \mathcal{S}_\mathcal{I} \setminus \mathcal{P}(\rho) \\ \tau \notin \{\tau_1, \tau_2\}}} \frac{\delta}{m! - |\mathcal{P}(\rho)| - 1}$$

$$= \underbrace{\sum_{\pi \in \mathcal{S}_\mathcal{I}} P(\Pi = \pi \mid X = x_i)}_{=1} + 2\delta - \delta - \delta$$

$$= 1,$$

where we used that the second sum has $|\mathcal{P}(\rho)| - 1$ many summands, since only either $\tau_1$ or $\tau_2$ are excluded, while the third sum has $m! - |\mathcal{P}(\rho)| - 1$ many summands, since $\mathcal{S}_\mathcal{I}$ has $m!$ many elements of which $|\mathcal{P}(\rho)|$ many are removed and in addition either $\tau_1$ or $\tau_2$ are excluded. Now it holds that $h$ is sub-k calibrated on $\mathcal{X} \setminus \mathcal{X}_0 \cup \mathcal{Z}$:

$$P(\Pi_{||\mathcal{B}} = \rho \mid h_{||\mathcal{P}}(X) = q)$$

$$= \sum_{\substack{x \in \mathcal{X} \setminus \mathcal{X}_0 \cup \mathcal{Z} \\ h_{||\mathcal{P}}(x) = q}} P(\Pi_{||\mathcal{B}} = \rho \mid X = x) \, P(X = x \mid h_{||\mathcal{P}}(X) = q) \tag{41}$$

$$= \sum_{\substack{x \in \mathcal{X} \setminus \mathcal{X}_0 \\ h_{||\mathcal{P}}(x) = q}} P(\Pi_{||\mathcal{B}} = \rho \mid X = x) \, P(X = x \mid h_{||\mathcal{P}}(X) = q)$$

$$+ \sum_{\substack{i \in [n] \\ h_{||\mathcal{P}}(x) = q}} P(\Pi_{||\mathcal{B}} = \rho \mid X = z_i) \, P(X = z_i \mid h_{||\mathcal{P}}(X) = q) \tag{42}$$

$$= \sum_{\substack{x \in \mathcal{X} \setminus \mathcal{X}_0 \\ h_{||\mathcal{P}}(x) = q}} P(\Pi_{||\mathcal{B}} = \rho \mid X = x) \, P(X = x \mid h_{||\mathcal{P}}(X) = q)$$

$$+ \sum_{\substack{i \in [n] \\ h_{||\mathcal{P}}(x)=q}} P(X = z_i \mid h_{||\mathcal{P}}(X) = q) \cdot P(\Pi_{||\mathcal{B}} = \rho \mid X = z_i) \tag{43}$$

$$= \sum_{\substack{x \in \mathcal{X} \\ h_{||\mathcal{P}}(x)=q}} P(\Pi_{||\mathcal{B}} = \rho \mid X = x) \, P(X = x \mid h_{||\mathcal{P}}(X) = q)$$

$$+ \sum_{\substack{i \in [n] \\ h_{||\mathcal{P}}(x)=q}} P(X = z_i \mid h_{||\mathcal{P}}(X) = q) \cdot \sum_{\pi \in \mathcal{P}(\rho)} \begin{cases} \delta & \text{if } \pi \in \{\tau_1, \tau_2\} \\ -\frac{\delta}{|\mathcal{P}(\rho)|-1} & \text{if } \pi \in \mathcal{P}(\rho) \setminus \{\tau_1, \tau_2\} \end{cases} \tag{44}$$

$$= q[\rho] + \sum_{\substack{i \in [n] \\ h_{||\mathcal{P}}(x)=q}} P(X = z_i \mid h_{||\mathcal{P}}(X) = q) \cdot \sum_{\pi \in \mathcal{P}(\rho)} \begin{cases} \delta & \text{if } \pi \in \{\tau_1, \tau_2\} \\ -\frac{\delta}{|\mathcal{P}(\rho)|-1} & \text{if } \pi \in \mathcal{P}(\rho) \setminus \{\tau_1, \tau_2\} \end{cases} \tag{45}$$

$$= q[\rho] + \sum_{\substack{i \in [n] \\ h_{||\mathcal{P}}(x)=q}} P(X = z_i \mid h_{||\mathcal{P}}(X) = q) \cdot \left( \delta - \frac{\delta}{|\mathcal{P}(\rho)| - 1}(|\mathcal{P}(\rho)| - 1) \right) \tag{46}$$

$$= q[\rho]. \tag{47}$$

Here, (41) is by the law of total probability. (42) is by splitting the sum into the partition of the summands, while (43) uses (40). Using (39) parts of the sums can be merged as in (44). Since $h$ is full-rank calibrated on $X$ and therefore sub-k calibrated on $X$ we obtain (45), evaluating the sum leads to (46), while (47) holds as the last term in brackets is zero.

Now it holds that $h$ is not top-k calibrated on $\mathcal{X} \setminus \mathcal{X}_0 \cup \mathcal{Z}$. Indeed, following the same steps as before and exchanging $\mathcal{P}$ by $\mathcal{T}$ we obtain:

$$P(\Pi_{||\mathcal{B}} = \rho \mid h_{||\mathcal{T}}(X) = q)$$

$$= \sum_{\substack{x \in \mathcal{X} \\ h_{||\mathcal{T}}(x)=q}} P(\Pi_{||\mathcal{B}} = \rho \mid X = x) \, P(X = x \mid h_{||\mathcal{T}}(X) = q)$$

$$+ \sum_{\substack{i \in [n] \\ h_{||\mathcal{T}}(x)=q}} P(X = z_i \mid h_{||\mathcal{T}}(X) = q) \cdot \sum_{\pi \in \mathcal{T}(\rho)} \begin{cases} \delta & \text{if } \pi \in \{\tau_1, \tau_2\} \\ -\frac{\delta}{|\mathcal{P}(\rho)|-1} & \text{if } \pi \in \mathcal{P}(\rho) \setminus \{\tau_1, \tau_2\} \end{cases} \tag{48}$$

$$= q[\rho] + \sum_{\substack{i \in [n] \\ h_{||\mathcal{T}}(x)=q}} P(X = z_i \mid h_{||\mathcal{T}}(X) = q) \cdot \sum_{\pi \in \mathcal{T}(\rho)} \begin{cases} \delta & \text{if } \pi \in \{\tau_1, \tau_2\} \\ -\frac{\delta}{|\mathcal{P}(\rho)|-1} & \text{if } \pi \in \mathcal{P}(\rho) \setminus \{\tau_1, \tau_2\} \end{cases} \tag{49}$$

$$= q[\rho] + \left( \delta - \frac{\delta}{|\mathcal{P}(\rho)| - 1}(|\mathcal{T}(\rho)| - 1) \right) \tag{50}$$

$$\neq q[\rho]. \tag{51}$$

Here, (48) is by the same arguments as for obtaining (44). Note that $\mathcal{T}(\rho) \subset \mathcal{P}(\rho)$, so the otherwise case in (40) does not appear. (49) is since $h$ is full-rank calibrated on $\mathcal{X}$, while (50) evaluates the sum. Finally, (50) holds as $|\mathcal{T}(\rho)| < |\mathcal{P}(\rho)|$.

$\square$

**Theorem A.3.** *Assume that the number of items $m \geq 3$. Then it holds:*

$$\forall k < m - 1 : \ \exists h \in \mathbb{P}(\mathcal{S}_\mathcal{I}) : h \text{ top-k calibrated } \wedge h \text{ not sub-k calibrated}.$$

*For $k = m - 1$ it holds that top calibration induces sub-k calibration.*

*Proof of Theorem A.3.* The proof is analogous to Theorem A.2, exchanging the roles of $\mathcal{P}$ and $\mathcal{T}$. For the case of $k = m-1$, it holds obviously using Corollary A.6 and then Theorem 4.2. $\square$

**Corollary A.4.** *Assume that the number of items $m \geq 3$. Then it holds:*

$$\forall k \in \{2, \ldots, m-1\} \, \exists h \in \mathcal{X} \to \mathbb{P}(\mathcal{S}_{\mathcal{I}}) : h \text{ sub-k calibrated } \wedge h \text{ not full-rank calibrated}.$$

*Proof of Corollary A.4.* This is a direct consequence of Theorem A.2 and Theorem 4.2 (ii): We can find a model that is sub-k calibrated, but not top-k calibrated and consequently not full-rank calibrated. $\square$

Notice that for Plackett–Luce and Mallows Models the conditions on $h$ hold, such that the proof of Theorem A.4 is also valid for those.

**Corollary A.5.** *Assume that the number of items $m \geq 3$. Then it holds:*

$$\forall k \in \{1, \ldots, m-2\} \, \exists h \in \mathcal{X} \to \mathbb{P}(\mathcal{S}_{\mathcal{I}}) : h \text{ top-k calibrated } \wedge h \text{ not full-rank calibrated}.$$

*Proof of Corollary A.5.* This is a direct consequence of Theorem A.3 and Theorem 4.2 (i): We can find a model that is top-k calibrated, but not sub-k calibrated and consequently not full-rank calibrated. $\square$

**Corollary A.6.** *If h is top-(m-1) calibrated, then h is full-rank calibrated*

*Proof of Corollary A.6.* Recall that the marginalization of top-k marginalization (Definition 3) uses exactly $c = (m-k)!$ many rankings $\pi \in \mathcal{S}_{\mathcal{I}}$ where $k \in \{1, \ldots, m\}$. In particular for $k = (m-1)$ we have exactly one ranking $\pi$ in the marginalisation, as $c = (m - (m-1))! = 1$. Obviously, it must hold that $h \in \mathcal{H}$ being top-$(m-1)$ calibrated also induces $h$ full-rank calibrated. $\square$

**Theorem A.7.** *Under the same assumptions as Theorem A.1 it holds:*

$$\forall k \in \{2, \ldots, m-1\} \, \exists h \in \mathcal{X} \to \mathbb{P}(\mathcal{S}_{\mathcal{I}}) : h \text{ rankwise calibrated } \wedge h \text{ not sub-k calibrated}.$$

*Proof of Theorem A.7.* We proceed similarly to the proof of Theorem A.1: Let $\mathcal{X}$ consist without loss of generality of $n < \infty$ data points for which $h \in \mathcal{H}$ be rankwise and sub-k calibrated for some $k \in \{2, \ldots, m\}$ arbitrary but fixed. Let $\mathcal{B} \subseteq \mathcal{I}$ be a subset of items with $|\mathcal{B}| = k$, $\pi, \tau \in \mathcal{S}_{\mathcal{I}}, q \in \mathbb{P}(\mathcal{S}_{\mathcal{B}})$ and $\rho \in \mathcal{S}_{\mathcal{B}}$ arbitrary but fixed such that $\pi \in \mathcal{P}(\rho)$. Additionally, we choose $q$ such that $q[\rho] \neq \alpha$. We choose the data points in $\mathcal{X}$ such that there exist two $x_i, x_j \in \mathcal{X}$ for which $x_i, x_j \in \mathcal{G} = \{x \mid h(x)[\pi] = \alpha\}$, $x_i \in \mathcal{F} = \{x \mid h_{||\mathcal{P}}(x) = q\}$, $x_j \notin \mathcal{F}$ and $P(X = x_i \mid h(x_i)[\pi] = \alpha) = P(X = x_j \mid h(x_j)[\pi] = \alpha)$. Intuitively, $\mathcal{G}$ are the data points for which $h$ assigns the probability $\alpha$ to the ranking $\pi$ and $\mathcal{F}$ are the points for which the marginals $h_{||\mathcal{P}}$ are identical to the probability vector $q$. Importantly, we have chosen $x_i, x_j$ in such a way, that both are assigned the same probability for $\pi$, i.e., $h(x_i)[\pi] = h(x_j)[\pi] = \alpha$ but their marginals are not identical, i.e., $h_{||\mathcal{P}}(x_i) \neq h_{||\mathcal{P}}(x_j)$.

From here we can proceed similarly to the proof of Theorem A.1 and construct the same $z_1, \ldots, z_n$ based on $x_1, \ldots, x_n$ such that $h$ is rankwise calibrated on these yet not sub-k calibrated. The same reason applies here: Since $x_i, x_j$ are elements of $\mathcal{G}$ and we have that $P(X = x_i \mid h(x_i)[\pi] = \alpha) = P(X = x_j \mid h(x_j)[\pi] = \alpha)$ we can repeat the same steps (28) – (33) to show that $h$ is rankwise calibrated on $z_1, \ldots, z_n$. Since $x_i$ is an element of $\mathcal{F}$, but $x_j$ is not, $h$ is not sub-k calibrated on $z_1, \ldots, z_n$.

$\square$

**Theorem A.8.** *Assume that the number of items $m \geq 3$ and let $\mathcal{X} = \{X_1, \ldots, X_n\}$ be the set of (random) data points for which $n < \infty$. Then it holds:*

$$\forall k \in \{1, \ldots, m-2\} \, \exists h \in \mathcal{X} \to \mathbb{P}(\mathcal{S}_{\mathcal{I}}) : h \text{ rankwise calibrated } \wedge h \text{ not top-k calibrated}.$$

*Proof of Theorem A.8.* The proof is analogous to Theorem A.7, by replacing $\mathcal{P}$ for $\mathcal{T}$. $\square$

**Theorem A.9.** *Assume that the number of items $m \geq 3$. Then it holds:*

$$\forall k < m : \, \exists h \in \mathcal{X} \to \mathbb{P}(\mathcal{S}_{\mathcal{I}}) : h \text{ rankwise sub-k calibrated } \wedge h \text{ not rankwise top-k calibrated}.$$

*Proof of Theorem A.9.* The proof is quite similar to the proof of Theorem A.2, where instead of

1. choosing some $q \in \mathbb{P}(\mathcal{S}_\mathcal{B})$ arbitrary but fixed, take some $\alpha \in [0,1]$ arbitrary but fixed,

2. choosing $h$ to be full-rank calibrated, use an $h$ that is rankwise calibrated,

3. conditioning on $h_{||\mathcal{P}}(X) = q$, condition on $h_{||\mathcal{P}}(X)[\rho] = \alpha$,

4. conditioning on $h_{||\mathcal{T}}(X) = q$, condition on $h_{||\mathcal{T}}(X)[\rho] = \alpha$,

5. replace each appearance of $q[\rho]$ by $\alpha$.

$\square$

**Theorem A.10.** *Assume that the number of items $m \geq 3$. Then it holds:*

$$\forall k < m : \exists h \in \mathcal{X} \to \mathbb{P}(\mathcal{S}_\mathcal{I}) : h \text{ rankwise top-k calibrated } \wedge h \text{ not rankwise sub-k calibrated}.$$

*Proof of Theorem A.10.* This proof is analogous to Theorem A.9, replacing $\mathcal{P}$ for $\mathcal{T}$. $\square$

**Corollary A.11.** *Assume that the number of items $m \geq 3$. Then it holds:*

$$\forall k \in \{2, \ldots, m-1\} \exists h \in \mathcal{X} \to \mathbb{P}(\mathcal{S}_\mathcal{I}) : h \text{ rankwise sub-k calibrated } \wedge h \text{ not rankwise calibrated}.$$

With a very similar proof technique one derives the following corollaries.

**Corollary A.12.** *Assume that the number of items $m \geq 3$. Then it holds:*

$$\forall k \in \{1, \ldots, m-2\} \exists h \in \mathcal{X} \to \mathbb{P}(\mathcal{S}_\mathcal{I}) : h \text{ rankwise top-k calibrated } \wedge h \text{ not rankwise calibrated}.$$

Notice that Theorems A.7, A.8, A.13 and A.14 also apply to Mallows Models and Plackett–Luce models, as there is no condition imposed on $h$ other than being calibrated.

**Theorem A.13.** *Let $\mathcal{X}$ consist of at least two elements. If there exist two data points $x_i, x_j \in \mathcal{X}$ with $i \neq j$ such that $P(X = x_i \mid h(x_i)[\alpha] = \alpha) \neq P(X = x_j \mid h(x_j)[\alpha] = \alpha)$ for some $\alpha \in [0,1]$, then it holds:*

$$\forall k \in \{2, \ldots, m-1\} \exists h \in \mathcal{X} \to \mathbb{P}(\mathcal{S}_\mathcal{I}) : h \text{ sub-k calibrated } \wedge h \text{ not rankwise calibrated}.$$

*Proof of Theorem A.13.* We proceed similarly to the proofs of Theorem A.1 and Theorem A.7: Let $\mathcal{X}$ consist without loss of generality of $n < \infty$ data points for which $h \in \mathcal{H}$ be rankwise and sub-k calibrated for some $k \in \{2, \ldots, m\}$ arbitrary but fixed. Let $\mathcal{B} \subseteq \mathcal{I}$ be a subset of items with $|\mathcal{B}| = k$, $\pi, \tau \in \mathcal{S}_\mathcal{I}, q \in \mathbb{P}(\mathcal{S}_\mathcal{B})$ and $\rho \in \mathcal{S}_\mathcal{B}$ arbitrary but fixed such that $\pi \in \mathcal{P}(\rho)$. Additionally, we choose $q$ such that $q[\rho] \neq \alpha$. We choose the data points in $\mathcal{X}$ such that there exist two $x_i, x_j \in \mathcal{X}$ for which $x_i, x_j \in \mathcal{F} = \{x \mid h_{||\mathcal{P}}(x) = q\}$, $x_i \in \mathcal{G} = \{x \mid h(x)[\pi] = \alpha\}$ and $x_j \notin \mathcal{G}$. Intuitively, $\mathcal{G}$ is the set of data points for which $h$ assigns the probability $\alpha$ to the ranking $\pi$, and $\mathcal{F}$ is the set of points for which the marginals $h_{||\mathcal{P}}$ are identical to the probability vector $q$. Importantly, we have chosen $x_i, x_j$ in such a way that they have different probability for $\pi$, i.e., $h(x_i)[\pi] \neq h(x_j)[\pi] = \alpha$ but their marginals are identical, i.e., $h_{||\mathcal{P}}(x_i) = h_{||\mathcal{P}}(x_j)$.

From here we can proceed similarly to the proofs of Theorem A.1 and Theorem A.7. That is, we construct the same $z_1, \ldots, z_n$ based on $x_1, \ldots, x_n$ such that $h$ is sub-k calibrated on these yet not rankwise calibrated. The same reason applies here: Since $x_i, x_j$ are elements of $\mathcal{F}$ and we have that we can repeat the same steps (28) – (33) by using the appropriate marginalizations and projections to show that $h$ is sub-k calibrated on $z_1, \ldots, z_n$. Since $x_i$ is an element of $\mathcal{G}$, but $x_j$ is not, and $P(X = x_i \mid h(x_i)[\pi] = \alpha) \neq P(X = x_j \mid h(x_j)[\pi] = \alpha)$, we see that $h$ is not rankwise calibrated on $z_1, \ldots, z_n$.

$\square$

**Theorem A.14.** *Assume that the number of items $m \geq 3$. Then it holds:*

$$\forall k \in \{1, \ldots, m-2\} \exists h \in \mathcal{X} \to \mathbb{P}(\mathcal{S}_\mathcal{I}) : h \text{ top-k calibrated } \wedge h \text{ not rankwise calibrated}.$$

*Proof of Theorem A.14.* The proof is analogous to Theorem A.13, replacing $\mathcal{P}$ for $\mathcal{T}$. $\square$

With a very similar proof technique one derives the following corollaries.

**Corollary A.15.** *Assume that the number of items $m \geq 3$. Then it holds:*

$$\forall k \in \{2, \ldots, m-1\} \, \exists h \in \mathcal{X} \to \mathbb{P}(\mathcal{S}_\mathcal{I}) : h \text{ rankwise sub-k calibrated} \wedge h \text{ not sub-k calibrated}.$$

**Corollary A.16.** *Assume that the number of items $m \geq 3$. Then it holds:*

$$\forall k \in \{1, \ldots, m-2\} \, \exists h \in \mathcal{X} \to \mathbb{P}(\mathcal{S}_\mathcal{I}) : h \text{ rankwise top-k calibrated} \wedge h \text{ not top-k calibrated}.$$

## B. Datasets Details

| Name | #of instances | #of features | #of items | #of rankings | fraction |
|---|---|---|---|---|---|
| authorship | 841 | 70 | 4 | 17 | 0.7083 |
| glass | 214 | 9 | 6 | 30 | 0.0417 |
| iris | 150 | 4 | 3 | 5 | 0.8333 |
| libras | 360 | 90 | 15 | 317 | $< 0.001$ |
| vehicle | 846 | 18 | 4 | 18 | 0.7500 |
| vowel | 528 | 10 | 11 | 294 | $< 0.001$ |
| wine | 178 | 13 | 3 | 5 | 0.8333 |
| yeast | 1484 | 8 | 10 | 471 | $< 0.001$ |
| **movies** | 602 | 65 | 15 | 260 | $< 0.001$ |
| **political** | 170 | 46 | 6 | 87 | 0.1208 |

*Table 4.* Datasets used for the experiments. The movies and political datasets correspond to real-world problems.

All datasets used in our experiments are publicly available on OpenML. An overview of the datasets employed for benchmarking is provided in Table 4, where *political* and *movies* constitute real-world datasets. The other datasets are adapted versions of classical classification tasks to the problem of label ranking Hüllermeier et al. (2008). Specifically, for each dataset, we first train a Naive Bayes classifier. The predicted class probabilities are then used to induce a ranking over all labels by sorting them in descending order. In the event of ties, labels with smaller indices are ranked first.

The *political* dataset (Thies et al., 2024) is derived from Likert-scale survey questions, such as "How conservative would you rate CDU/CSU?", which serve as features. The learning task is to predict an overall ranking of the political parties *LINKE*, *GRUENE*, *SPD*, *CDU/CSU*, *FDP*, and *AfD*. The *movies* dataset (Harper & Konstan, 2016) consists of a subset of 602 instances and contains ratings for the 15 most frequently rated movies. These movies are ranked according to their assigned star ratings, following the construction described by Alfaro et al. (2023a). All data files used in our experiments are available in our public repository: `https://github.com/Advueu963/Calibrated_Preference_Learning`.

## C. Implementation Details

The ranking models are implemented in PyTorch (Paszke et al., 2019), using neural networks with two hidden layers of size 100 and ReLU activations. For the Plackett–Luce model and RankClassifier, we use the Adam optimizer (Kingma & Ba, 2015) with learning rate $10^{-3}$ and no weight decay. On each dataset, we train for 50 epochs using a batch size of 64.

**RankClassifier** RankClassifier is a multi-class classification model that outputs a probability distribution over all possible rankings $\mathcal{S}_{\mathcal{I}}$. Naively training such a model is infeasible for large $m$, since the number of rankings grows factorially as $m!$. Following Resin (2023), we therefore learn a distribution only over the $N$ unique rankings observed in the training data and assign uniform probability mass to all remaining rankings. Concretely, let $x$ denote the total probability mass assigned to the $N$ observed rankings. All other rankings are assigned probability $\frac{1-x}{m!-N}$. We implement this by modifying the softmax normalization to implicitly account for $m! - N$ additional logits that share a constant value $c$ (instead of explicitly representing $m! - N$ outputs). Let $z_r$ denote the logit for an observed ranking $r$. Then the probability of an observed ranking is

$$P(r) \;=\; \frac{e^{z_r}}{\sum_{r' \in \mathcal{R}_{\text{obs}}} e^{z_{r'}} + (m! - N)e^c},$$

and each unobserved ranking receives probability

$$P(r_{\text{unobs}}) \;=\; \frac{e^c}{\sum_{r' \in \mathcal{R}_{\text{obs}}} e^{z_{r'}} + (m! - N)e^c}.$$

This is memory efficient and allows us to represent full ranking distributions even for moderately large item sets (e.g., $m = 15$). Assigning uniform probability mass to all unobserved rankings can be interpreted as a prior assumption; in contrast, assigning very negative logits to these rankings corresponds to assuming that such rankings cannot occur. Model parameters are learned by minimizing the cross-entropy loss between the predicted distribution and the observed ranking.

**Plackett–Luce model** The Plackett–Luce model is learned by minimizing the negative log-likelihood of the observed rankings as described in (Cheng et al., 2010). The learned logits are exponentiated and normalized to obtain the corresponding choice probabilities. Since the Plackett–Luce model is not uniquely identified without a normalization constraint (Vitelli et al., 2017), we normalize the parameters accordingly. To avoid numerical issues, we subtract the maximum logit prior to exponentiation.

**Mallows model** The Mallows model is learned in interleaved steps, since both the dispersion parameter $\lambda$ and the reference ranking $\tau$ must be estimated. Following Vitelli et al. (2017), we initialize $\tau$ as an "average" ranking in the training data by computing each item's average position and sorting items by these averages (ties are broken randomly). We then alternate between updating $\lambda$ using the closed-form solution in Vitelli et al. (2017) and updating $\tau$ via the local search procedure of Busse et al. (2007).

**Ranking by pairwise comparisons (RPC)** Ranking by pairwise comparisons is implemented using the approach of Alfaro et al. (2023a). We first construct a pairwise comparison dataset from the rankings by considering all implied pairwise preferences. Then, for each item pair, we train a binary classifier to predict which item is preferred. We use scikit-learn's `DecisionTreeClassifier`, as it has been reported to perform well in this setting (Hüllermeier & Furnkranz, 2004). We apply Platt scaling (Platt et al., 1999) as a post-hoc calibration method to ensure that the underlying binary models output calibrated probabilities. Finally, we aggregate the pairwise predictions using the Borda method (McLean et al., 1995) to obtain a ranking over all items. We base our implementation on Alfaro et al. (2021) and adapt it to the label ranking setting. Importantly the calibration of RPC can not be investigated for anything else than sub-2, as it inherently only learns sub-2 pairwise probabilities.

**Plackett–Luce-RPC** For RPC one cannot directly obtain a probability distribution over all rankings, since the model only yields pairwise preference probabilities. Interpreting these probabilities as Bradley–Terry probabilities (Bradley & Terry, 1952), we use the (improved) Zermelo algorithm (Newman, 2023) to obtain Plackett–Luce parameters, as described in Section 3. Given $p_{i,j} = P(i \succ j)$ for all $i, j \in \mathcal{I}$, we iteratively update parameters $\theta_i$ for $t = 1, \ldots, T$ via

$$\theta_i^{(t+1)} \;=\; \frac{\sum_{j \neq i} \frac{p_{i,j}\,\theta_i^{(t)}}{\theta_i^{(t)} + \theta_j^{(t)}}}{\sum_{j \neq i} \frac{p_{j,i}}{\theta_i^{(t)} + \theta_j^{(t)}}} \,.$$

As shown empirically by Newman (2023), this update converges faster than the original Zermelo algorithm (Zermelo, 1929) while yielding the same estimates. We set $T = 100$, which is safely above typical convergence thresholds (e.g., Newman (2023) report that fewer than 20 iterations can already achieve a squared difference of $10^{-6}$). This procedure relates to rank-breaking, where pairwise preferences are used to infer a distribution over full rankings (Soufiani et al., 2013).

# D. Reward Model Calibration

This appendix details the reward model calibration experiments presented in Section 5.2.

## D.1. Setup

**Dataset.** RewardBench2 (Malik et al., 2025) evaluates reward models across six categories: Factuality, Precise Instruction Following, Math, Safety, Focus, and Ties. Each datapoint is a six-tuple $(Q, R_1, R_2, R_3, R_4)$, where $Q$ is the prompt, $R_1, R_2, R_3, R_4$ are candidate responses, and $T \in \{1, 2, 3, 4\}$ indicates the correct answer. We exclude the "Ties" category, which allows for more than four responses with multiple correct answers and thus does not fit our top-1 formulation.

**Model selection.** We evaluate all Bradley–Terry-trained reward models from the RewardBench2 leaderboard and report the results for the ten best-calibrated ones in Figure 3a. We exclude two models (`nicolinho/QRM-Gemma-2-27B` and `nicolinho/QRM-Llama3.1-8B-v2` by Dorka (2024)) trained with quantile regression, as their outputs cannot be interpreted as Plackett–Luce utilities.

**Label ranking formulation.** Although ranking free-form responses is naturally an object ranking problem, with the responses themselves having attributes beyond their identity (Kamishima et al., 2011), the fixed four-response structure of RewardBench2 allows for a label ranking formulation. We define the feature space $\mathcal{X}$ as the set of prompt-response tuples $(Q, R_1, R_2, R_3, R_4)$ and take labels $\mathcal{I} = \{1, 2, 3, 4\}$ to represent response positions in this context. The prediction task is then to identify which position contains the best response.

## D.2. Calibration Measurement

Given a reward model $r$, we define the induced top-1 prediction model by

$$h_r\big((Q, R_1, R_2, R_3, R_4)\big)[i] = \frac{r(Q, R_i)}{\sum_{j=1}^{4} r(Q, R_j)} \,.$$

This is the top-1 probability under the Plackett–Luce model with utilities given by the reward scores.

**ECE computation.** For each context, the model predicts a top-1 probability for each of the four positions. We compute ECE by pooling all (context, position) predictions, binning by predicted probability, and comparing mean predicted probability to empirical accuracy within each bin. We use 10 equal-width bins.

# E. Calibration in Label Ranking

## E.1. Evaluation Metrics: Critical View and Discussion

Expected Calibration Error (ECE) is a widely adopted measure for assessing the miscalibration of probabilistic classifiers. However, beyond its well-known sensitivity to binning hyperparameters, ECE exhibits limited expressiveness as the number of classes increases. As a consequence, several alternative calibration measures have been proposed, most of which aim to reduce the effective number of observed classes (Nixon et al., 2019). This limitation becomes particularly severe in the context of label ranking, where the number of possible classes grows factorial and already exceeds $100$ for as few as $5$ items.

As the number of items increases, probabilistic ranking models inevitably predict increasingly flat distributions over rankings. In this regime, the absolute differences used in ECE shrink in magnitude, leading to ECE values that are close to zero, almost independent of the actual calibration quality. We mitigate this issue by restricting the computation of ECE to the $95\%$ most frequent rankings observed in the dataset. For instance, in the *movies* dataset, only $260$ distinct rankings are observed in the data, despite there being $15!$ possible rankings in total. Among these observed rankings, we retain the $95\%$ most frequent ones based on empirical frequency. Nevertheless, even under this restriction, ECE remains close to zero, as shown in Figure 6.

These observations indicate that using absolute differences in ECE becomes increasingly intractable for large-scale label-ranking problems. They also raise the more fundamental question of whether predicting complete distributions over rankings is meaningful for large item sets such as $m = 15$, which we do not address in this work. One possible direction to alleviate this limitation is to replace absolute error measures with relative divergence-based metrics. For example, the Kullback–Leibler divergence $D_{KL}$, or the symmetric Jeffreys divergence

$$D_{\text{jeff}}(p\|q) = D_{KL}(p\|q) + D_{KL}(q\|p),$$

where $p$ denotes the empirical frequency and $q$ the predicted distribution, could be employed. Compared to $D_{KL}$, $D_{\text{jeff}}$ has the appealing property of detecting miscalibration even when the model underestimates the true probability mass. Figure 5 showcases the potential differences when using a relative measure, such as the Jeffreys divergence, in contrast to using the standard absolute difference.

Such alternatives require systematic investigation, which we leave to future work and instead propose a stagewise evaluation strategy. Despite low absolute ECE values, meaningful relative comparisons between models remain possible (e.g., determining whether one model is better calibrated than another). Moreover, to assess calibration for rankings of length $k$, we recommend starting with calibration at $k' = 2$. If ECE is high (e.g., above $0.1$), the model is clearly uncalibrated for larger rankings such as $k = 5$. If ECE is acceptable, calibration can then be examined progressively for $k' = 3$ and higher values. This strategy exploits the inherent calibration structure of label ranking models (see Figure 1), since calibration at higher $k$ necessarily implies calibration for all lower-order interactions.

To assess full-rank calibration, we employ the canonical multiclass calibration error estimator based on Dirichlet kernel density estimation proposed by Popordanoska et al. (2022). This estimator targets the strongest notion of multiclass calibration, often referred to as canonical or distribution calibration (Alvo & Philip, 2014), which requires the entire predicted probability vector over rankings to be calibrated. Intuitively, the method estimates the conditional empirical distribution of true rankings in a local neighborhood of each predicted probability vector, where locality is defined via a Dirichlet kernel. The calibration error is then computed as the average $\ell_2$ distance between the predicted probability vector and this locally estimated empirical distribution:

$$\text{Full-Rank-ECE} = \frac{1}{n} \sum_{j=1}^{n} \left\| \frac{\sum_{i \neq j} k_{Dir}(h(x_j); h(x_i)) y_i}{\sum_{i \neq j} k_{Dir}(h(x_j); h(x_i))} - h(x_j) \right\|_2^2$$

where

$$k_{Dir}(h(x_i); h(x_j)) = \frac{\Gamma(\sum_{k=1}^{K} \alpha_{i,k})}{\prod_{k=1}^{K} \Gamma(\alpha_{i,k})} \prod_{k=1}^{K} h(x_j)_k^{\alpha_{i,k}-1}$$

where $\alpha_{i,k} = \frac{h(x_i)_k}{S_i} + 1$, $S_i = \sum_{k=1}^{K} h(x_i)_k$ and $y_i$ is the number of $\pi_i$ when enumerating the rankings $\mathcal{S}_{\mathcal{I}}$. To apply this method we first encode the given (sub-) rankings into a classification problem, via enumeration of the rankings. For full rank sub-k or full rank top-k calibration, we use enumerate only the rankings present in the subset and not the entire set $\mathcal{S}_{\mathcal{I}}$.

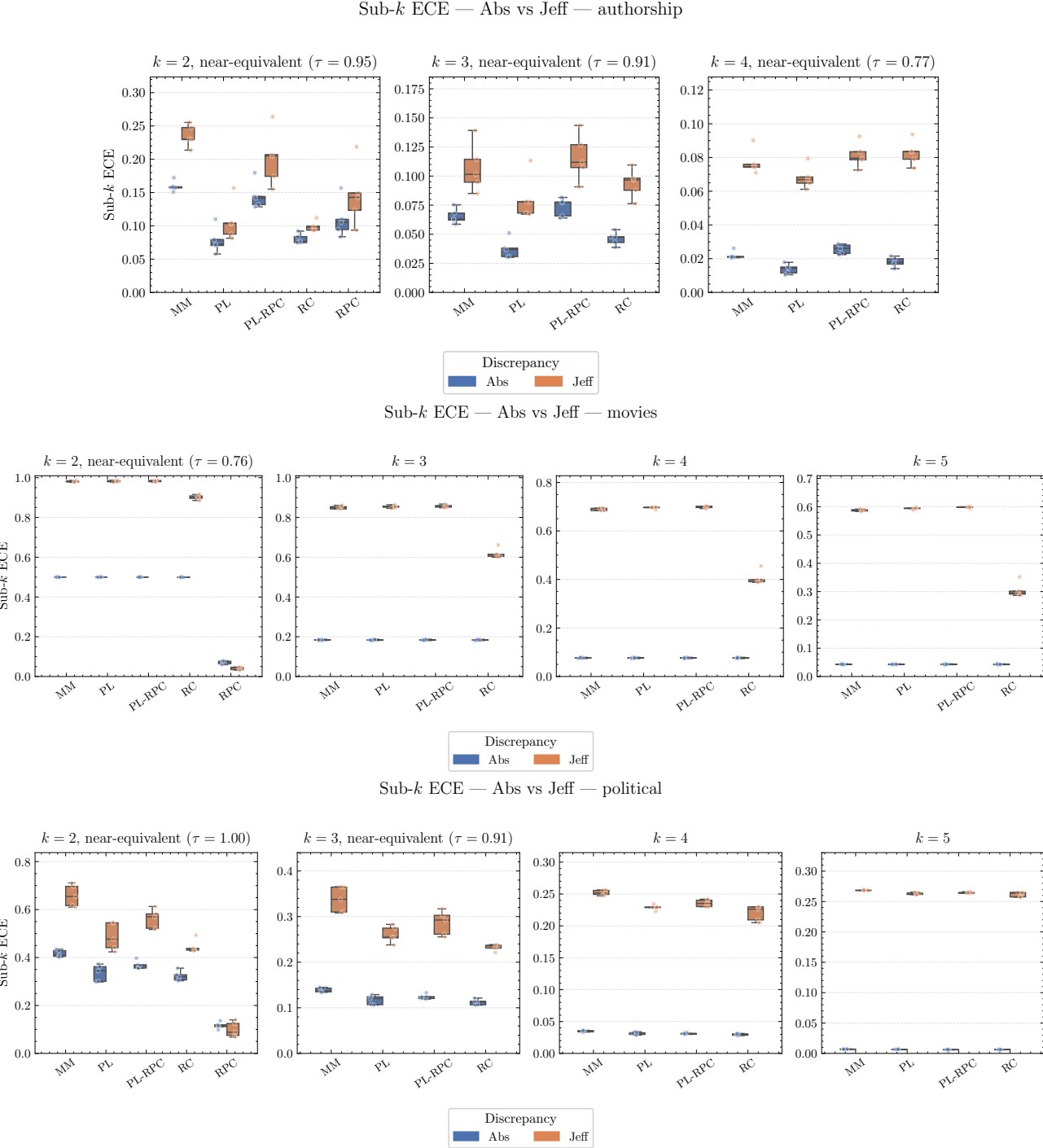

*Figure 5.* Comparison of using $|\cdot|$ in contrast to Jeffrey Divergence for calculating sub-k calibration via ECE. Rank correlation of the methods is reported only if the values using $|\cdot|$ are significantly different.

For example $2 \succ 1 \succ 3$, might obtain the number 3 when enumerating over $\mathcal{S}_{\mathcal{I}}$, yet if the consider the sub-ranking $1 \succ 3$ it obtains the number 2. Unlike binning-based approaches, the Dirichlet kernel estimator provides a smooth and consistent estimate of full-distribution miscalibration in high-dimensional probability simplices, making it particularly well suited for evaluating calibration of probabilistic label ranking models over the full ranking space.

## E.2. Further Results

We report complete results for both full-rank / rankwise sub-$k$ calibration and top-$k$ calibration of the considered label ranking models. Full-rank calibration is computed using the Dirichlet kernel proposed by Popordanoska et al. (2022). Overall, models tend to be well calibrated on smaller datasets such as *iris* (Figure 7) and *authorship* (Figure 8), but become increasingly uncalibrated on datasets with larger ranking spaces, such as *movies* (Figure 6) and *political* (Figure 9). Across nearly all datasets, RPC and RankClassifier are among the best calibrated models, as both can leverage standard calibration methods for classification (Aly, 2005). RPC is most often the best calibrated model, while RankClassifier performs best on *iris* (Figure 7), *authorship* (Figure 8), *vehicle* (Figure 10), and *wine* (Figure 13). In contrast, the Mallows model consistently exhibits the worst calibration, since it learns only global parameters, which may fail to capture instance-specific ranking uncertainty. Among native label ranking models, Plackett–Luce achieves the strongest calibration performance, as its weights are learned individually for each data point rather than globally. Nevertheless, as illustrated on *vowel* (Figure 12), *yeast* (Figure 14), and *glass* (Figure 15), Plackett–Luce can still be substantially uncalibrated. Consistent with rank breaking effects, the Plackett–Luce–RPC variant shows inferior calibration, as it is trained only on pairwise comparisons but predicts a distribution over full rankings. We further observe that, despite low ECE values for larger item sets ($k > 4$), almost all models exhibit high ECE for $k = 2$ and $k = 3$. This behavior is explained by the rapidly increasing number of possible rankings for $k \geq 4$, which leads to extremely small probability masses and consequently small values of $|\text{acc}(I_b) - \text{conf}(I_b)|$, artificially reducing ECE. This suggests that more sensitive calibration measures should be explored, which we leave for future work. Overall, these results highlight the need for calibration methods tailored to label ranking, as (i) RankClassifier incurs high computational cost and (ii) RPC only produces distributions over pairwise comparisons, which do not reliably generalize to full ranking distributions.

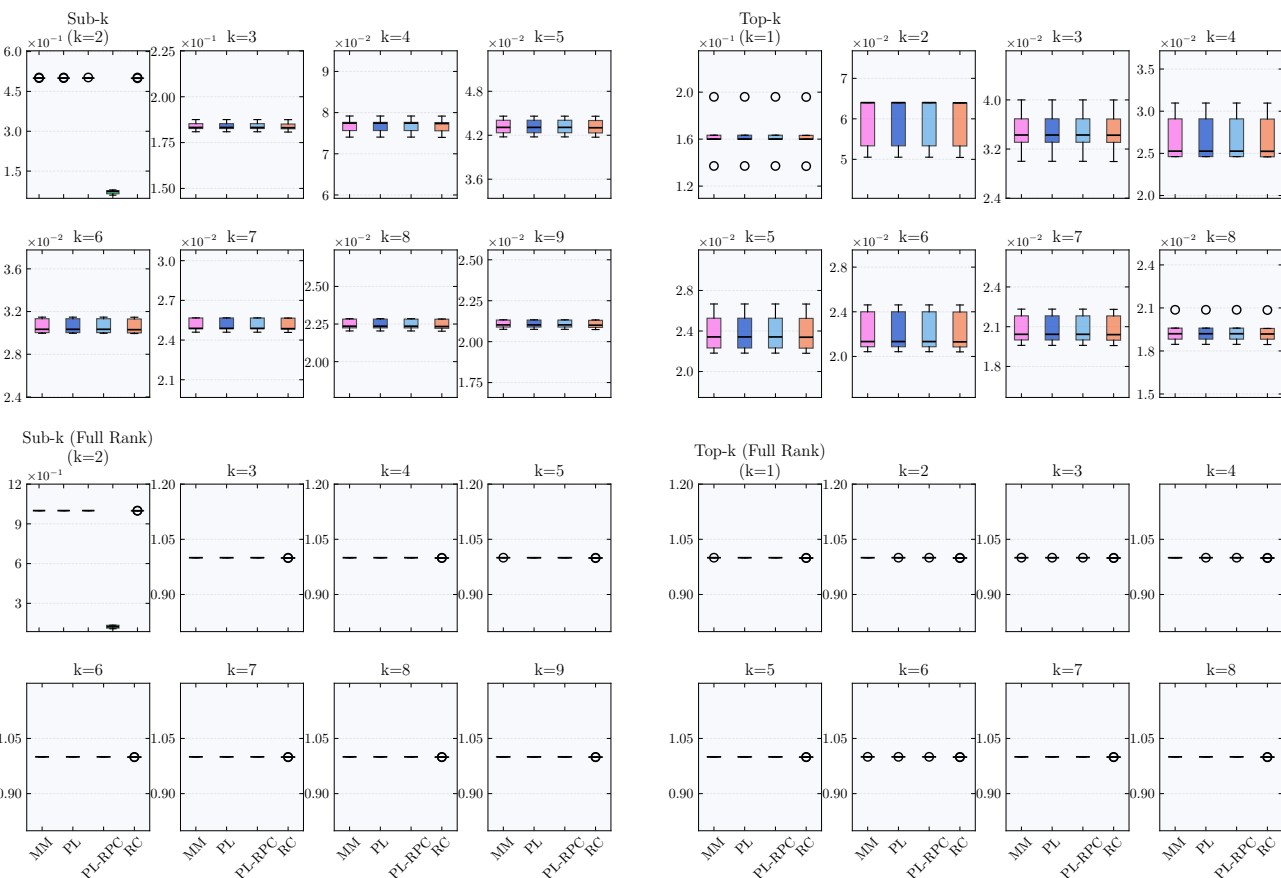

*Figure 6.* Calibration of label ranking models for the "movies" dataset, considering only the rankings covering 95% of the probability mass.

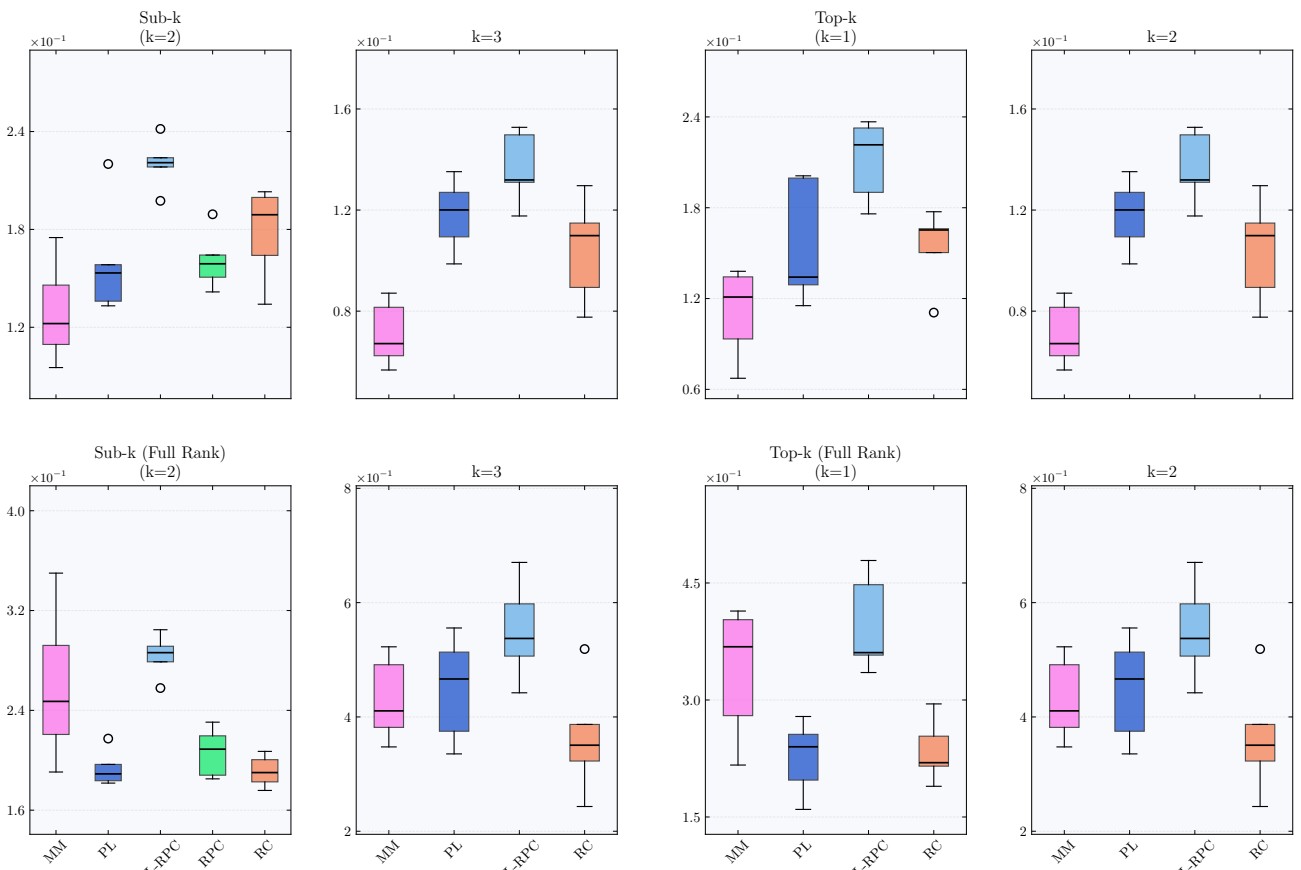

*Figure 7.* Calibration of label ranking models for the "iris" dataset, considering only the rankings covering 95% of the probability mass.

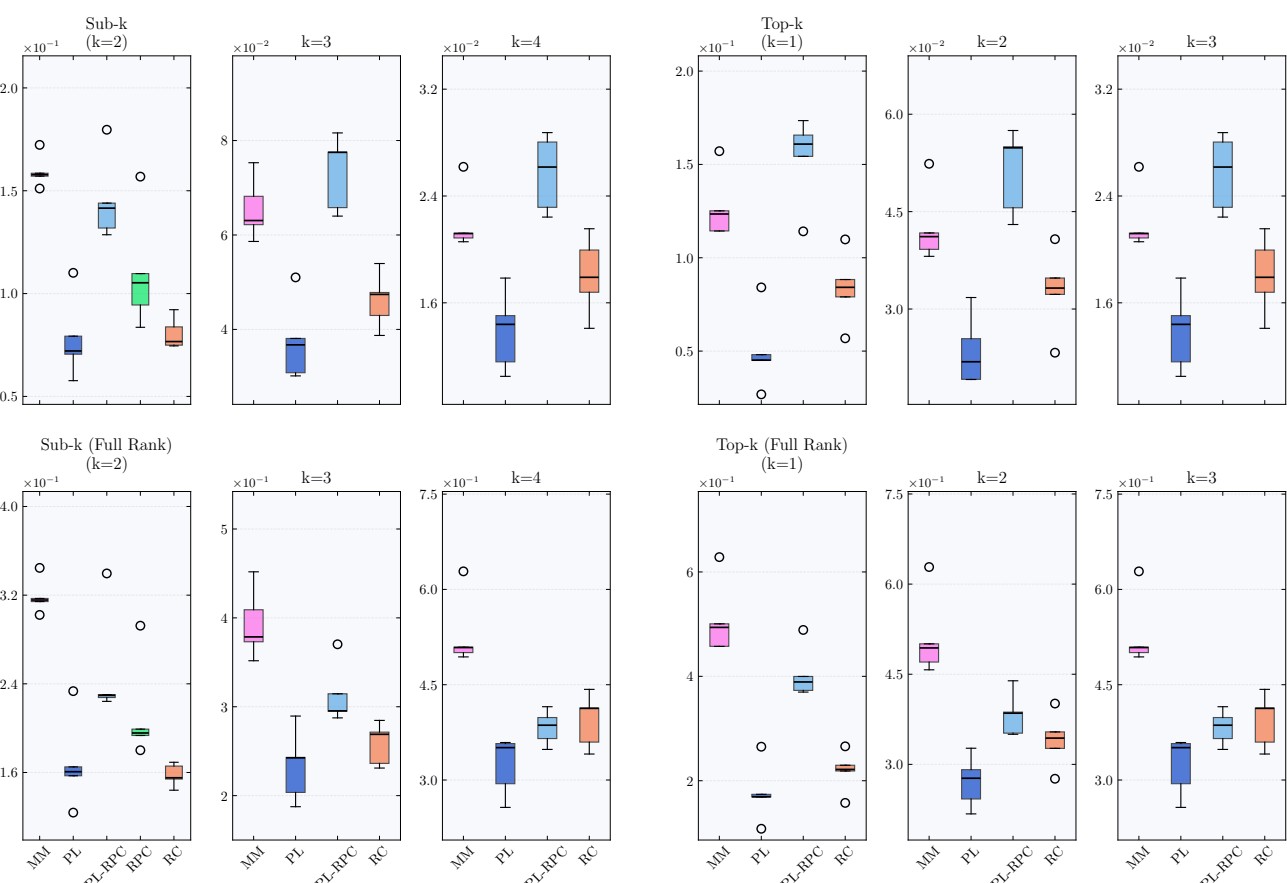

*Figure 8.* Calibration of label ranking models for the "authorship" dataset, considering only the rankings covering 95% of the probability mass.

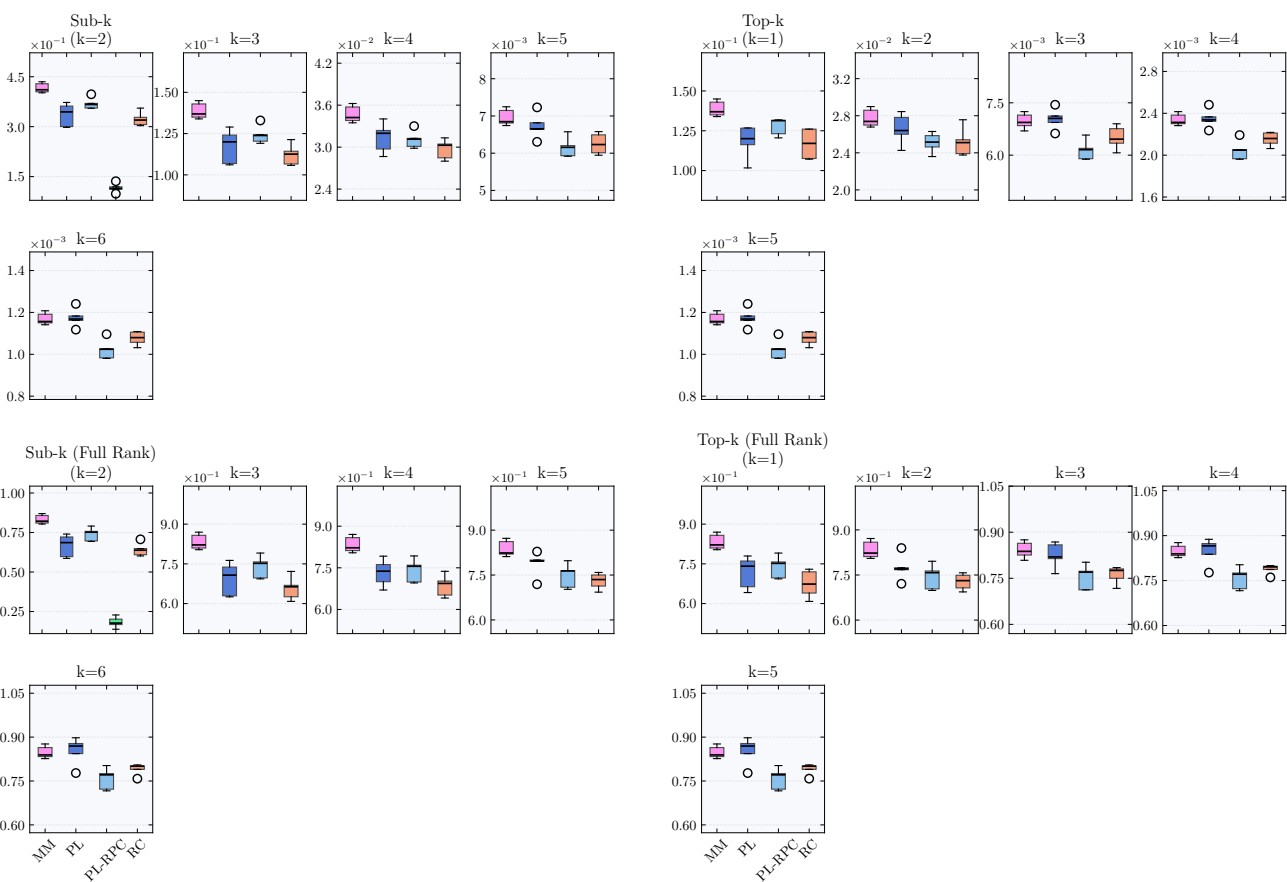

*Figure 9.* Calibration of label ranking models for the "political" dataset, considering only the rankings covering 95% of the probability mass.

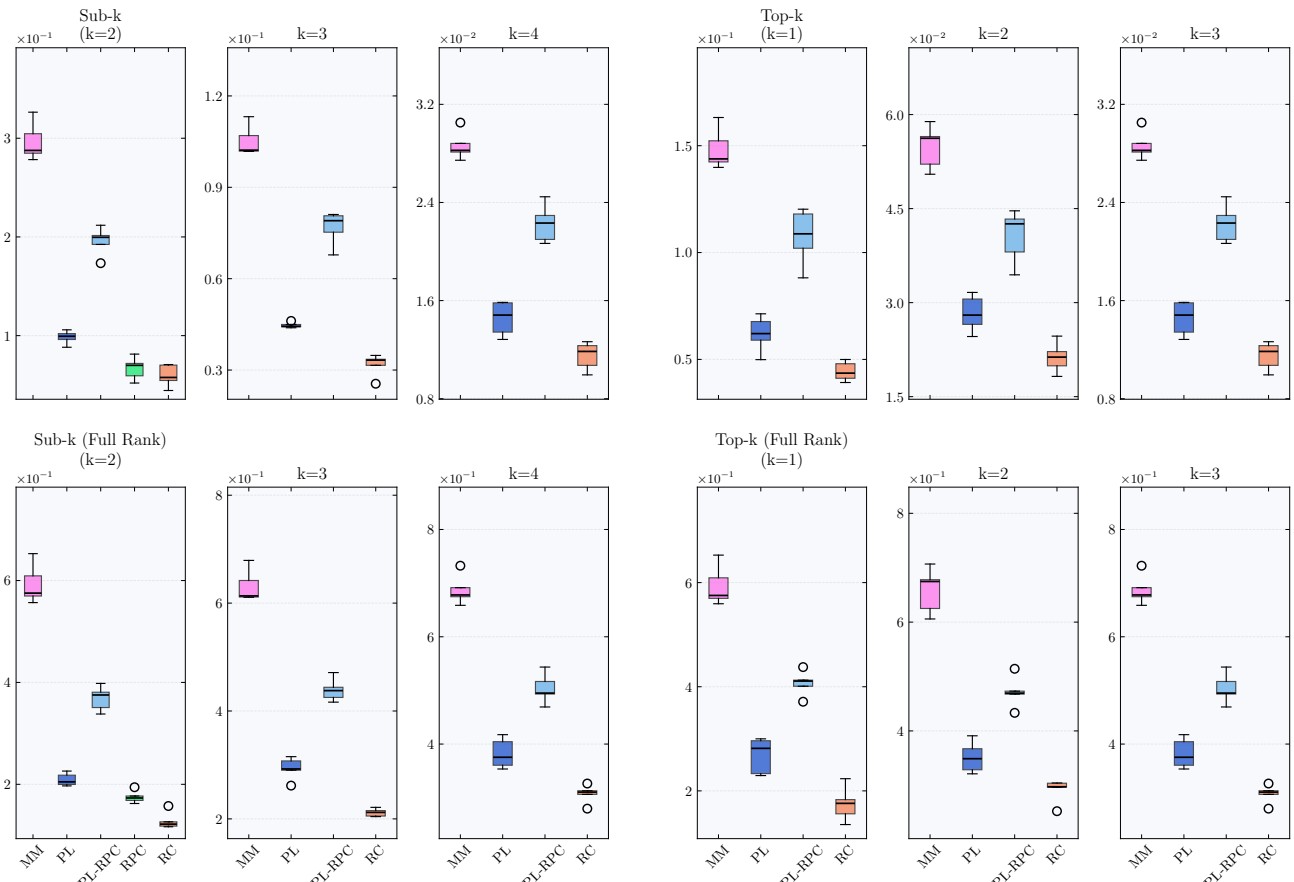

*Figure 10.* Calibration of label ranking models for the "vehicle" dataset, considering only the rankings covering 95% of the probability mass.

ECE calibrations on segment (sub-k 2..9, top-k 1..8)

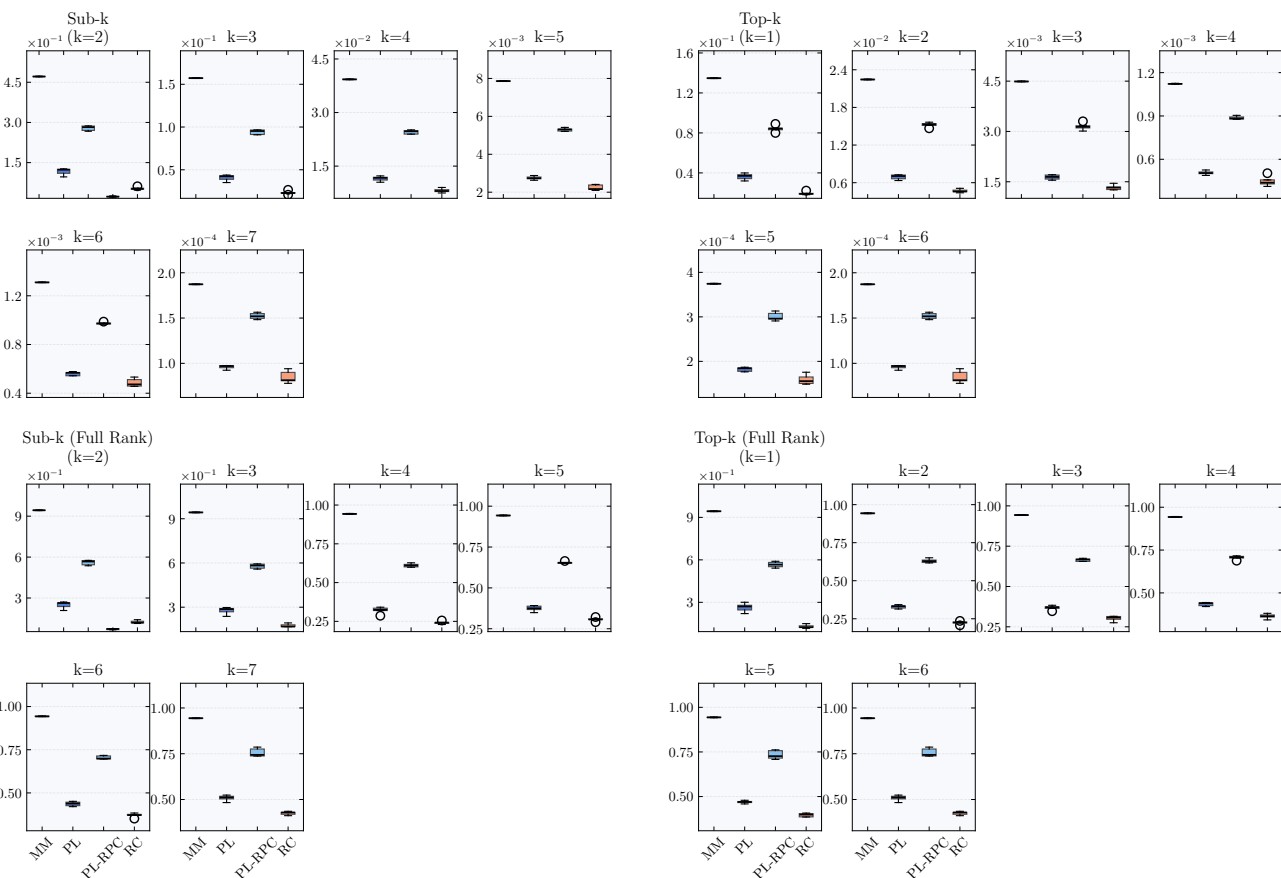

*Figure 11.* Calibration of label ranking models for the "segment" dataset, considering only the rankings covering 95% of the probability mass.

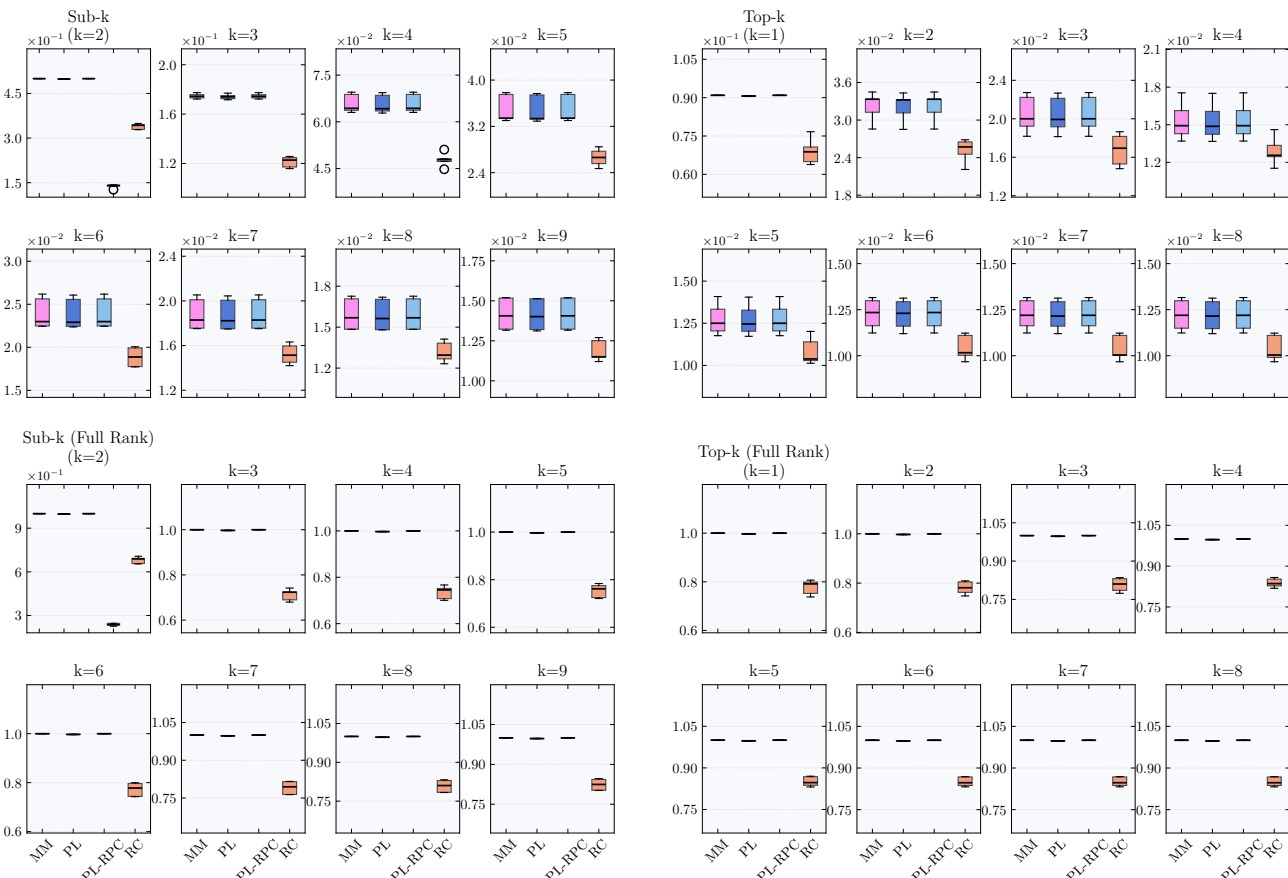

*Figure 12.* Calibration of label ranking models for the "vowel" dataset, considering only the rankings covering 95% of the probability mass.

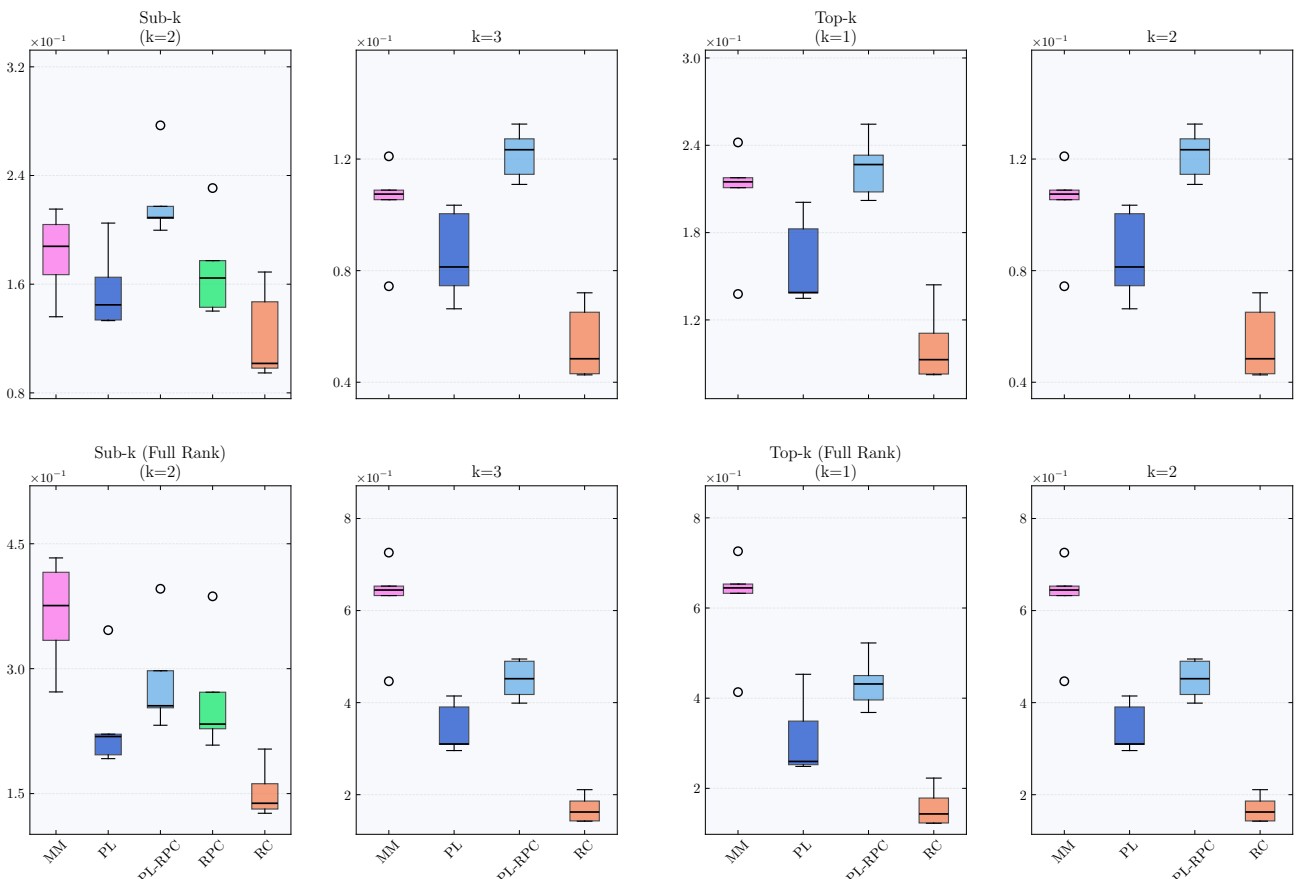

*Figure 13.* Calibration of label ranking models for the "wine" dataset, considering only the rankings covering 95% of the probability mass.

ECE calibrations on yeast (sub-k 2..9, top-k 1..8)

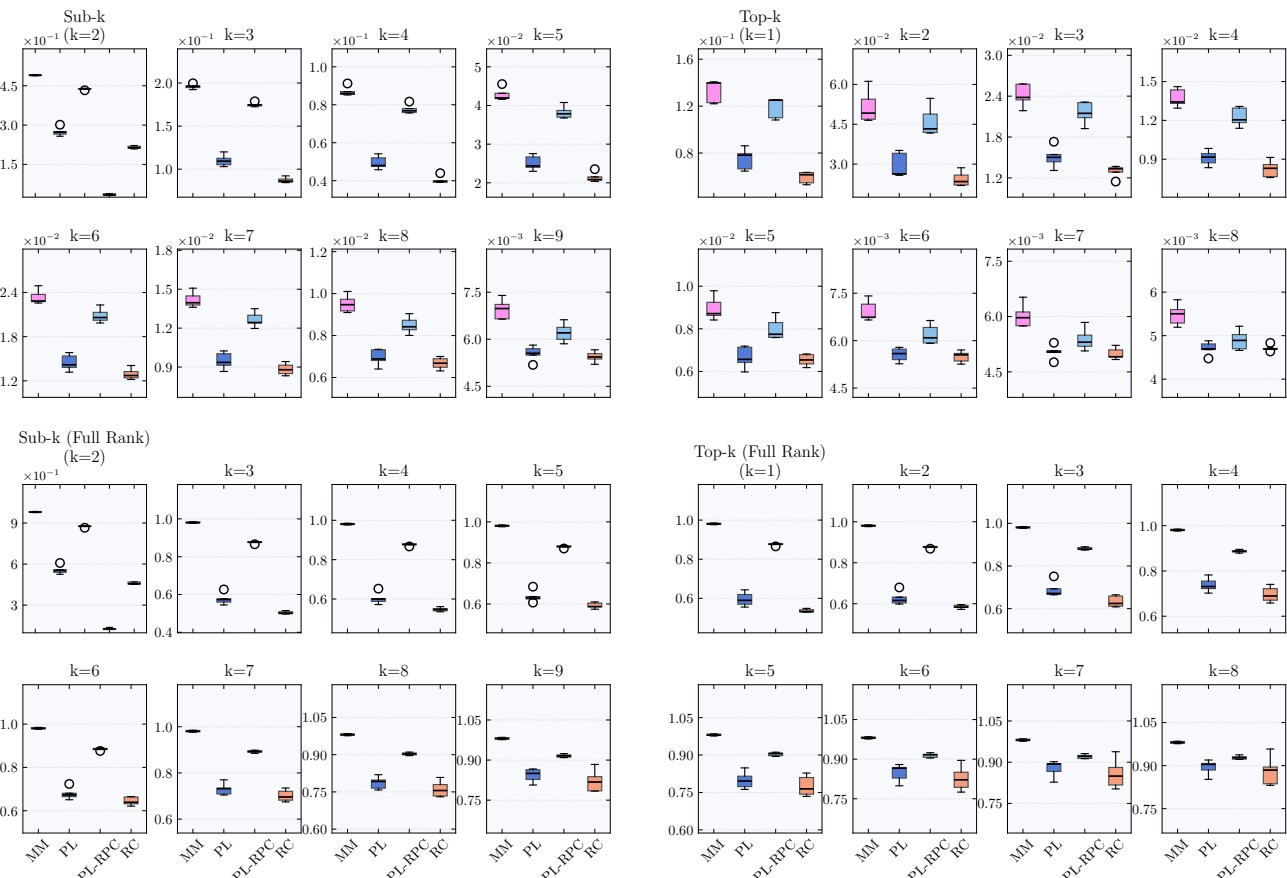

*Figure 14.* Calibration of label ranking models for the "yeast" dataset, considering only the rankings covering 95% of the probability mass.

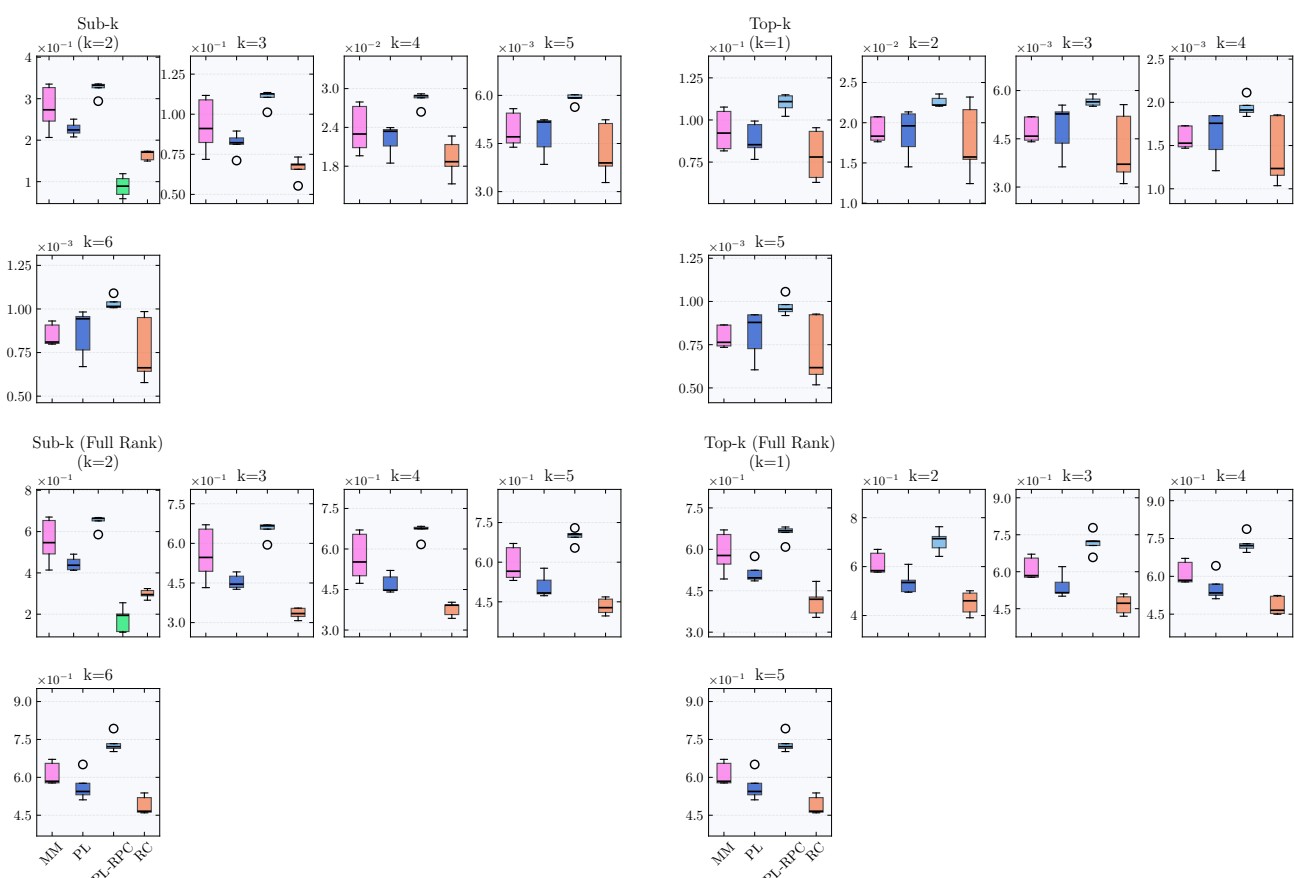

*Figure 15.* Calibration of label ranking models for the "glass" dataset, considering only the rankings covering 95% of the probability mass.

