# OpenReview forum: "Calibrated Preference Learning: The Case of Label Ranking"
_ICML.cc/2026/Conference — ICML 2026 regular_

### Official Review · Reviewer_AsMN · 2026-02-23

**Soundness:** 3
**Presentation:** 3
**Significance:** 3
**Originality:** 3
**Overall Recommendation:** 4
**Confidence:** 3

**Summary:**

This paper introduces a calibration framework for probabilistic label ranking with theoretical proving that full-rank calibration implies sub-ranking and top-k calibration but not vice versa.

**Compliance With Llm Reviewing Policy:**

Affirmed.

**Final Justification:**

Thanks for the author's detailed explanation. However, due to my lack of expertise in the calibrated learnng area, I maintain my initial positive rating after rebuttal.

**Key Questions For Authors:**

- Given that ECE systematically approaches 0 as the number of items grows, is there any alternative metric can meaningfully measure calibration in large ranking spaces?

**Limitations:**

The paper does not contain an explicit limitation section, only Appendix E.1 mentions that the ECE evaluation metric is ill-suited for large ranking spaces. Also, the RLHF experiment is limited in scope, relying on a dataset with only 4 candidate responses and evaluating only top-1 calibration, leaving pairwise calibration (the most relevant modality for RLHF) untested.

**Strengths And Weaknesses:**

**Strengths:**

- The proposed calibration framework for probabilistic label ranking is solid with theoretical guarantees and is complete. Also, the derivation is standard and straightforward.
- The paper addresses a very practical problem and fills the gap of calibration in label ranking.


**Weakness:**
- The ECE evaluation metric is fundamentally ill-suited for this setting (the authors themselves also noted this in Appendix E.1). The proposed mitigation strategy of restricting to the top 95\% most frequent rankings does not resolve this.

- The RLHF experiment is not very convincing. RewardBench2 only contains 4 candidate responses per prompt, this is a scale at which standard classification calibration would suffice. Therefore, the advantages of ranking-specific framework are not demonstrable here.

---

> ### Author Rebuttal · Authors · 2026-03-30
>
> We appreciate your comments on our paper. The concern around ECE’s limitations is fair, and we address each concern below.
>
> > **W1**: The ECE evaluation metric is fundamentally ill-suited for this setting (the authors themselves also noted this in Appendix E.1). The proposed mitigation strategy of restricting to the top 95% most frequent rankings does not resolve this.
>
> As we discuss in Appendix E.1, ECE is not ideal for large ranking spaces. Since our **primary contribution is theoretical** (defining calibration notions for label ranking and their relationships), the experiments serve as a proof of concept, and ECE remains standard and interpretable for this purpose. Importantly, while absolute ECE values shrink in large ranking spaces, relative comparisons between models remain meaningful, which is what our experiments focus on.
>
> More broadly, **measuring calibration when the outcome space is very large is a research question in its own right, and one we consider out of scope for this work**. To illustrate the difficulty: one natural idea is to replace absolute error with a divergence that uses log-ratios, such as KL or its symmetric variant, Jeffrey divergence. Log-ratios remain sensitive to miscalibration even as distributions flatten and probabilities become small (as opposed to absolute error). [These figures](https://anonymous.4open.science/r/Calibrated_Preference_Learning/Rebuttal_Figure.pdf) show ECE (left) and Jeffrey divergence (right); our conclusions hold across both metrics.
>
> However, even **Jeffrey divergence does not fully resolve the problem**. In large ranking spaces, an averaged metric such as ECE or the Jeffrey variant would otherwise be dominated by a large number of unobserved rankings on which the model may appear well-calibrated. We therefore still restrict both metrics to the 95% most frequent rankings in the figures linked above. The challenge is not just about picking the right metric but about evaluation in combinatorially large outcome spaces more generally.
>
>
> > **W2**: The RLHF experiment is not very convincing. RewardBench2 only contains 4 candidate responses per prompt, this is a scale at which standard classification calibration would suffice. Therefore, the advantages of ranking-specific framework are not demonstrable here.
>
> There are two reasons why standard full-rank calibration is not the right tool here, even with only 4 candidates. First, and most directly, the **ground-truth full rankings are not available in this task**; we only observe the top-1 choice, so full-rank calibration is not applicable. Second, as our theoretical results show, a model may be top-1 calibrated but not full-rank calibrated. **When the downstream task is top-1 selection, applying full-rank calibration could therefore lead to overconservative estimates**, measuring calibration over parts of the ranking that are irrelevant to the task at hand.
> This is precisely the kind of distinction our framework is designed to make explicit: different choice tasks warrant different notions of calibration.
>
> > **Q1:** Given that ECE systematically approaches 0 as the number of items grows, is there any alternative metric can meaningfully measure calibration in large ranking spaces?
>
> As discussed in our response to W1, **Jeffrey divergence is a promising direction**: it does not suffer from the same collapse towards zero and produces results consistent with our ECE-based findings, though it still requires the 95% restriction to avoid being dominated by unobserved rankings. A full theoretical treatment of alternative metrics for ranking calibration is a natural next step, but lies beyond the scope of the current paper, which focuses on establishing the notion of calibration for label ranking in the first place.

---

> > ### Author Rebuttal · Reviewer_AsMN · 2026-04-02
> >
> > Thanks for the response. After reading the answers to W1, I have a follow-up question, so given the inherent difficulties in evaluation, how would practitioners know whether their ranking model is well-calibrated in real-world settings? I noticed that the other reviewer also raised concerns about practical significance, so I would expect the authors to provide a justification in this regard.

---

> > > ### Author Response · Authors · 2026-04-02
> > >
> > > Thank you for responding. We address your question below and hope to clarify your remaining concerns.
> > >
> > > > How would practitioners know whether their ranking model is well-calibrated in real-world settings?
> > >
> > > This is a fair question. The inherent difficulties of evaluation stem from the problem of having too many rankings to consider. Please note, however, that achieving full calibration of all these rankings is almost impossible in practice anyway and, perhaps more importantly, not actually of true interest. It is difficult to envisage a scenario in which a practitioner would be concerned about whether the predicted probability for an extremely improbable ranking of, say, ten items is 0.00001 or 0.00002.
> > >
> > > This is a key motivation for the main contribution of our paper. Proceeding from full calibration as an ideal reference, we introduce various weaker notions of calibration, such as sub-k and top-k calibration. These are not only tractable but also arguably more relevant in practice. Please note that our motivation is fully aligned with that for concepts such as class-wise calibration and confidence calibration, which weaken the notion of strong calibration in the multi-class classification setting: in practice, achieving strong calibration is almost impossible, and the accuracy of the high-probability classes is much more important than that of the low-probability classes.
> > >
> > > Concretely, to compute the ECE, we suggest restricting to the calibration notion of interest (e.g., sub-2, top-2,..), and, if still intractable, to further constrain the set of relevant rankings (e.g., using the 95% most probable or the m-most frequent rankings). This provides a tractable way to compute practically relevant mis-calibration in ranking models.
> > >
> > > We also refer to Appendix E.1, where we propose a bottom-to-top approach: first investigating the calibration notion on weaker, more tractable variants, such as sub-2, and then considering more complex ones. This enables a practitioner to identify to what degree calibration still holds and at which $k$ it might break down.
> > >
> > > Triggered by your question, we will extend the discussion already present in Appendix E.1 to provide practitioners with a guide to measuring the calibration of ranking models in a tractable way.

---

### Official Review · Reviewer_atrk · 2026-03-07

**Soundness:** 2
**Presentation:** 2
**Significance:** 2
**Originality:** 2
**Overall Recommendation:** 4
**Confidence:** 3

**Summary:**

This paper studies calibration for **probabilistic label ranking**, where a model outputs a distribution $ h(x)\in \mathcal{P}(S_I) $
over permutations (rankings) of  items. Extending standard calibration templates from multi-class classification, the authors define calibration at multiple granularities: (i) **full-rank** calibration  (ii) **rankwise** calibration conditioning on a single probability value , and (iii) structure-aware notions based on induced marginals, including **sub-$k$** calibration (relative order within a subset) and **top-$k$** calibration (the top $k$ positions). The paper proves implication of all these calibration relationships (e.g., strong $\Rightarrow$ weak) . Experiments checks the calibration behaviour of common probabilistic ranking models (e.g., Plackett–Luce, Mallows) and pairwise-preference.

Overall, the taxonomy is conceptually helpful, but has very limited methodological novelty. The concepts of weak and strong calibration is already existing. The pope also lacks sufficient practical guidance for estimating these notions at scale, and limited actionable insight for improving calibration in real ranking systems.

**Compliance With Llm Reviewing Policy:**

Affirmed.

**Final Justification:**

I have read the rebuttal, as well as the concerns raised by other reviewers and the authors’ responses. While I remain unconvinced by several aspects of the paper, particularly the limitations I previously highlighted and the somewhat vague responses to those concerns, I acknowledge that other reviewers find the work compelling. Taking this into account, I am updating my score to a weak accept.

**Key Questions For Authors:**

1. **Estimators for strong calibration:** How should practitioners estimate $ \mathbb{P}(\Pi=\pi \mid h(X)=p)=p[\pi]$
   when exact repeats of $p$ are unlikely?
2. **Scalable evaluation for sub-$k$/top-$k$:** For moderate/large $m$, evaluating all subsets  is infeasible. What do you recommend in this settings, and how should uncertainty and coverage be reported?
3. **Matric-aligned targets:** Many ranking decisions depend on notions such as "item $I$ is in top-$k$” (ignoring within-top order) or pairwise win probabilities. Can these be incorporated as in calibration notions, and how do they relate to your hierarchy?
4. **Novelty:** Beyond diagnosing miscalibration, what calibration-improvement method do you recommend for different ranking models ?

**Limitations:**

- **Strong calibration is difficult to validate empirically.** Conditioning on $h(X)=p$ is theoretically clean but practically problematic; approximations can be unstable.
- **Combinatorial explosion.** Both $|S_I|=m!$ and the subset space for sub-$k$/top-$k$ hinder scalability.
- **Dependence on fully observed rankings.** Direct applicability is limited when supervision is partial or implicit.
- **Taxonomy without prescriptive guidance.** When notions are incomparable, the paper does not clearly prescribe which notion to prioritise for a given downstream decision context.

**Strengths And Weaknesses:**

## Strengths

- **Clear formal setup for ProLR.** Notation for rankings, subset restrictions, and induced marginals is consistent and supports later definitions.
- **Structure-aware calibration definitions.** Sub-$k$ and top-$k$ notions reflect realistic downstream needs beyond full permutations.
- **Helpful theoretical clarification.** Implication results and explicit counterexamples reduce confusion about which calibration guarantees transfer across modalities.
- **Relevance to preference learning / RLHF.** Treating pairwise as a key special case is practically motivated.

## Weaknesses

- **Limited novelty in core strong/weak notions.** Full-rank and rankwise calibration largely reuse standard calibration templates by viewing each $\pi\in S_I$ as a discrete outcome.
- **Scalability concerns remain unresolved.** The space $|S_I|=m!$ and the subset explosion for sub-$k$/top-$k$ make exhaustive evaluation infeasible; practical evaluation protocols are not clearly established.
- **Limited actionability.** The work is largely diagnostic/taxonomic and provides little concrete guidance on how to improve calibration in practical pipelines.
- **Potential mismatch with realistic supervision.** Many applications observe partial rankings or implicit feedback; it is not fully clarified how the framework extends without strong additional assumptions.

---

> ### Author Rebuttal · Authors · 2026-03-30
>
> We greatly appreciate your feedback and the time taken for this review. We hope our additional comments sufficiently address your concerns.
>
> > [W1]: Limited novelty in core strong/weak notions. Full-rank and rankwise calibration largely reuse standard calibration templates by viewing each as a discrete outcome.
>
> We acknowledge this in the introduction (line 44, right-hand side), as these are not our main contributions. The core novelty lies in extending the notion of calibration beyond what can be captured by viewing label ranking as a multi-class classification problem.
>
> > [W2]: Scalability concerns remain unresolved. The space and the subset explosion for sub-/top- make exhaustive evaluation infeasible; practical evaluation protocols are not clearly established.
>
> We disagree: for practically relevant values of $k$, it holds:
> - for sub-k, where $k=2$, complexity is $O(m^2)$.
> - for top-k, where $k=1$, complexity is $O(m)$.
>
> This is the complexity found in downstream tasks. For the other non-standard cases, we have an entire section (Appendix E.)  discussing exactly the case of practical evaluation protocols.
>
> > [W3]: Limited actionability. The work is largely diagnostic/taxonomic and provides little concrete guidance on how to improve calibration in practical pipelines.
>
> Yes, that's due to the clearly defined scope of our work. We provide a taxonomic overview and then diagnose the practical methods based on our concepts. That is already a highly non-trivial task, as demonstrated by nearly seven pages of proof, and above all, is a natural first step. Proposing methods to improve calibration is a logical next step, but deserves its own article, similar to classification (Kull et al., NeurIPS'19; Wenger et al., AISTATS'20).
>
> > [W4]: Potential mismatch with realistic supervision. Many applications observe partial rankings or implicit feedback; it is not fully clarified how the framework extends without strong additional assumptions.
>
> Our frameworks already consider partial rankings in the sense of observing only a ranking on a smaller set of items (see sub-k and top-k rankings and their calibration notions). Extending the framework to richer forms of implicit feedback is an interesting direction. However, this is heavily dependent on the specific downstream task and out of scope for this work.
>
> > [Q1]: Estimators for strong calibration: How should practitioners estimate $\mathbb{P}(\Pi=\pi \mid h(X)=p)=p[\pi]$ when exact repeats are unlikely?
>
> As we describe in Appendix E.1, we use the method of Popordanoska, et al. (NeurIPS'22), which is a tractable approximation of strong calibration.
>
> > [Q2]: Scalable evaluation for sub-/top-: For moderate/large $m$, evaluating all subsets is infeasible. What do you recommend in this settings, and how should uncertainty and coverage be reported?
>
> We would like to point out that in practice, one would not evaluate all combinations of subsets; instead, one is typically interested in specific cases, such as sub-2 or top-1. These do not have a severe scaling problem (see our answer to your W2).
>
> In general we would recommend considering only those rankings which are of highest interest. A principled way is to only consider those rankings, which occur in X% of the training data. Further, one can consider weaker calibration notions (sub-k/top-k calibration) which make the evaluation more tractable. Further discussion can be found in Appendix E.
>
> > [Q3]: Matric-aligned targets: Many ranking decisions depend on notions such as "item $I$ is top-$k$” (ignoring within-top order) or pairwise win probabilities. Can these be incorporated as in calibration notions, and how do they relate to your hierarchy?
>
> This is a really interesting question. Yes, it is straightforward to incorporate it into our framework: Define $R$ as the set of interesting rankings (i.e. those where $I$ is among the $k$ most preferred items). Then marginalisation of $h(X)[R]$ is the cumulative probability of rankings in $R$ (similarly as in Def. 3), and calibration as in Def. 5: $P(\Pi_{\mathcal{B}} = \rho \ | \ h(X)[R] = q) = q(\rho)$. It would then be a third branch in Figure 1 (and 2), implied by full rank calibration, and also by top-k calibration, but not sub-k. The latter can be shown by reducing it to a top-1 problem with m-(k-1) items.
>
> > [Q4]: Novelty: Beyond diagnosing miscalibration, what calibration-improvement method do you recommend for different ranking models ?
>
> One suitable calibration method is the RPC model, decomposing the ranking problem into sub-2 rankings problems. Each of the binary classifiers in RPC can be calibrated using methods such as Platt Scaling. However, this imposes a strong marginalisation, and as such building tailored methods for k>2 is non-trivial. As discussed in our response to W3, deriving calibration methods is a topic worthy of its own article, similar to classification (Kull et al., NeurIPS'19; Wenger et al., AISTATS'20).

---

> > ### Author Rebuttal · Reviewer_atrk · 2026-04-02
> >
> > Even after carefully considering the rebuttal, I remain unconvinced about the level of novelty and the practical significance of the proposed approach. While the authors have provided additional clarifications, the core contributions still appear incremental, and it is not entirely clear how the method would translate to meaningful real-world impact or broader applicability. That said, I acknowledge that other reviewers have raised complementary perspectives and highlighted strengths that I may have undervalued. Taking these viewpoints into account, and recognizing that the paper may still offer incremental progress of interest to the community, I am willing to revise my assessment.
> > As a result, I am willing to update my score to a weak accept, albeit with some reservations regarding the paper’s originality and practical relevance.

---

### Official Review · Reviewer_fzeh · 2026-03-12

**Soundness:** 3
**Presentation:** 2
**Significance:** 3
**Originality:** 3
**Overall Recommendation:** 4
**Confidence:** 3

**Summary:**

This paper studies the calibration of probabilistic label ranking (ProLR). The core idea is that since the model outputs a "probability distribution for all rankings," calibration should not only treat each complete ranking as a class, as in multi-class classification, but also consider the internal structure of the ranking, such as pairwise/sub-k/top-k. The authors also demonstrate the relationship between these relationships in detail, and experimental results support the conclusions to some extent.

**Compliance With Llm Reviewing Policy:**

Affirmed.

**Final Justification:**

Given the authors' response, I have changed my original score.

**Key Questions For Authors:**

See Weakness and limitation

**Limitations:**

The article presents itself as a highly theoretical work, but the notation, definitions, proofs, and citations don't match the density of its theoretical content. This leads to fatigue for me. The authors use numerous theorem/corollary/appendix proofs, which seem information-rich, but I constantly encounter issues such as notation reuse, unclear types, mixing of conditional events and random variables, and inconsistencies between the theorem statement and the proof's objects. I wonder, "If I've already found so many surface-level notation/rigor problems, how many more haven't I missed?"

**Strengths And Weaknesses:**

Strength

The question itself is novel. In label ranking/reward modeling, people often look at accuracy, pairwise correctness, and top-1 win rate, but rarely do they systematically ask, "Is your output ranking distribution calibrated or not?" This approach is reasonable. The hierarchical definition is insightful. It separates full ranking, sub-ranking, and top-k, and proves the relationship between full ranking and the other ranking methods

Weakness

I am a little confused about the definition in Eq.(3) and others as well. If I understand right, the formula here does not refer to the "conditional distribution of a given input X" in the usual sense, but rather to a calibration condition: given the predicted distribution reported by the model itself, the conditional distribution of the actual result should be consistent with this predicted distribution. However, if $h(X)$ is continuous, the event $\{h(X)=p\}$ should be 0, therefore the definition seems a little unrigorous (all the following proofs face the similar problem).

The manuscript’s proof contains many obscure and lengthy arguments. I carefully reviewed a portion of it and found some parts that I believe are problematic or not rigorous (perhaps due to time constraints, my understanding is not thorough enough). For instance, in Eq.(13), $p$ seems to be a fixed distribution rather than a random variable, therefore $p[\pi]=a$ is not an event. Moreover, in Eq.(14)-(16), $p(p[\pi]=a)$ is also incorrect because $p$ is not a random variable. In Line 807, the right-hand equation should not be $h(X)$ because it is a random variable?  In Line 856, the statement of $h$ is inconsistent. I remember it should a function rather than a distribution （such statements also appear in the following proofs). In the proof of Theorem A.1, the authors define the $p\in\mathcal{P}$ but use $q$ in the following statements. In Line 948, should $\max$ be $\min$? In Line 1042, Corollary A.4 states that "there exists a model that is sub-k calibrated but not full-rank calibrated," which is unrelated to the statement that "top-(m-1) induces sub-k.” How do the authors claim such a proof? Moreover, $P(A)$ is used simultaneously as both the "probability of the event" and the "set of distributions". Please check all the proofs, which make me exhausted when checking the correctness of the proofs and doubt the validity of the proof.

When the ranking space is large, the probability of a single complete ranking becomes very small, and the distribution becomes increasingly flat. In this case, even if the model is inaccurate, many absolute gaps will appear very small, so the ECE may approach 0. How do the authors evaluate the effectiveness of the proposed methods in such a condition.

---

> ### Author Rebuttal · Authors · 2026-03-30
>
> We appreciate your comments on our paper, and in particular the close reading of the proofs. We address each point below.
>
> > [W1]: Confusion about Eq. 3
>
> Eq. (3) follows the standard definition of calibration in classification. It is grounded in probability theory and a convention used in machine learning, not just in the context of calibration. The definitions are regular conditional probabilities, i.e., as equalities holding almost surely with respect to the distribution of $h(X)$ (or even a continuous $X$ as in a standard ML setting), as these are random variables themselves in the form of conditional expectations. Please let us know if there are any changes to the manuscript that could clarify this further.
>
> > [Q2.1]: "p in Eq.(13), seems to be a fixed distribution rather than a random variable, therefore is not an event p[\pi]=\alpha. Moreover, in Eq.(14)-(16), is also incorrect because it is not a random variable."
>
> We only partly agree. It is perfectly possible to condition on “fixed distributions” or rather constant random variables realizing some value. For example, define $Z=c$ for some fixed $c \in \mathbb{R}$. Then, for any $z  \neq c$ we have $\{Z=z\}= \emptyset$ and $\{Z=c\}= \Omega.$ Thus, the events are well-defined. Now replace $Z$ with $p[\pi]$ to get our setting in this theorem. The reviewer does identify a genuine issue in Eq. (14)-(16), since we need to integrate over $\{p:\, p[\pi]=\alpha\}$ and not sum over $p \in \mathbb{P},$ so that
>
>  $$ P(\Pi = \pi \\ | \\ h(X)[\pi] = \alpha) = \int_{\{p:\, p[\pi]=\alpha\}} P(\Pi = \pi \\ | \\ h(X) = p) \;dP(h(X)=p \\ | \\ h(X)[\pi]=\alpha)\,.$$
>
> Nevertheless, the argument of the proof is still the same. Similarly we have adjusted Eq. (25)-(27) to use $\int_{\{q:\, q[\tau]=\alpha\}}$ instead of the sum.
>
> > [Q2.2]: In Line 807, the right-hand equation should not be h(X) because it is a random variable?
>
> Yes, this is a typo. The right hand should be $q[\tau]$ as defined in Def. 4.
>
> > [Q2.3]: In Line 856, the statement of h is inconsistent.
>
> Thanks, yes this is again a typo: $h \in \mathcal{H}$ which is defined as $ \mathcal{X} \rightarrow \mathbb{P}(\mathcal{S}_I).$ We have corrected this typo here and in all other such statements.
>
> > [Q2.4]: In the proof of Theorem A.1, the authors define p \in \mathcal{P}  but use q in the following statements.
>
> We have addressed this typo. The new statement is $x_i \in \mathcal{F} = \{x \in \mathcal{X} \\ | \\ h(x) = p\}$, $x_j \notin \mathcal{F}$.
>
> > [Q2.5]: In Line 948, $max$ should be $min$ ?
>
> Yes, again a typo. The upper bound takes care that $P(\Pi = \pi \ | \ X = z_i)$ is a distribution, due to this the outer $max$ must be a $min$.
>
> > [Q2.6]: In Line 1042, Corollary A.4 states that "there exists a model that is sub-k calibrated but not full-rank calibrated," which is unrelated to the statement that "top-(m-1) induces sub-k.” How do the authors claim such a proof?
>
> We have not claimed that, but we see a possible source of the confusion. The proof of Corollary A.4 is using Theorem A.2 and not Theorem A.3. We guess that the reviewer is referring to the proof of Theorem A.3, where we had a wrong reference as it should have referred to Corollary A.6 and not A.4.
>
> > [Q2.7]: Moreover, $P(A)$ is used simultaneously as both the "probability of the event" and the "set of distributions". Please check all the proofs, which make me exhausted when checking the correctness of the proofs and doubt the validity of the proof.
>
> We do make a clear distinction: $P(A)$ is the probability of an event $A$ and $\mathbb{P}(\mathcal{A})$ is the set of distributions over a **set** $\mathcal{A},$ as stated in our **Notation** section. We have reviewed all proofs to confirm this distinction is maintained throughout.
>
> > [Q3]: "When the ranking space is large [...] ECE may approach 0. How do the authors evaluate the effectiveness of the proposed methods in such a condition."
>
> This is a valid concern. ECE itself is best suited for smaller problems such as the calibration for sub-k and top-k rankings. We discuss alternative metrics in Appendix E.1, and address this issue in more detail in our response to W1 and Q1 of reviewer AsMN.
>
> > [Q4]: "I constantly encounter issues such as notation reuse, unclear types, mixing of conditional events and random variables, and inconsistencies between the theorem statement and the proof's objects"
>
> We are very grateful to the reviewer for checking the proofs, as this is rarely done. Having addressed each point above, we want to emphasize that the remaining issues identified are typos and one minor notational correction in Eq. (14)-(16), none of which affect the proof arguments or the theoretical results. We have carefully revised all proofs and notation in the updated manuscript. We hope that our explanations above resolve the conceptual concerns, and that the remaining typos, which we have corrected in the updated manuscript, do not warrant rejection.

---

> > ### Author Rebuttal · Reviewer_fzeh · 2026-04-01
> >
> > I appreciate the authors' response. It really takes me lots of time to understand your work. Please double-check your proofs in case of any typos, which might confuse your readers. I decide to change my decision to weak accept.

---

### Official Review · Reviewer_57eJ · 2026-03-13

**Soundness:** 4
**Presentation:** 4
**Significance:** 4
**Originality:** 4
**Overall Recommendation:** 6
**Confidence:** 4

**Summary:**

This article studies a broad aspect of probabilistic label ranking, namely how to define and evaluate calibration for models that output distributions over rankings rather than single predictions. Overall, the authors study a core problem: whether probabilistic ranking models produce predictions whose probabilities are consistent with observed frequencies, and how such calibration can be formally defined and assessed. The paper introduces a hierarchy of calibration notions for ranking distributions (including full-rank, rankwise, sub-(k), and top-(k) calibration), analyzes their relationships theoretically, and proposes empirical procedures to evaluate them. The framework is applied to several ranking models and to reward models used in RLHF, demonstrating that commonly used models can exhibit significant calibration errors even when their ranking accuracy appears strong.

While the paper focuses on defining and analyzing calibration notions for probabilistic label ranking, the factorial growth of the ranking space makes full-rank calibration difficult to measure or estimate in practice as the number of items increases. The authors already note this challenge when discussing the limitations of ECE-based evaluation in large ranking spaces. One promising direction would be to explore approximate inference techniques for ranking distributions, which could make stronger calibration notions more tractable in practice. In particular, prior work on probabilistic inference over repeated insertion models (e.g., Kenig et al., AAAI 2018) develops scalable methods for reasoning about distributions over permutations. Such techniques may provide a useful starting point for developing practical calibration estimation or correction procedures for probabilistic ranking models.

**Compliance With Llm Reviewing Policy:**

Affirmed.

**Final Justification:**

Thank you for the rebuttal!  I was already happy with the paper before rebuttal, and am keeping my score of accept.

**Key Questions For Authors:**

N/A

**Limitations:**

The authors haven't discussed the limitations but them made their assumptions clear.

**Strengths And Weaknesses:**

The paper has several notable strengths.

S1. Important and (surprisingly) underexplored problem. The paper studies calibration for probabilistic label ranking, a topic that has received far less attention than calibration in classification or regression. The problem is both theoretically interesting and practically relevant, particularly in settings such as recommender systems and RLHF reward modeling where ranking distributions play a central role.

S2. Clean and well-motivated formulation. The authors introduce a clear hierarchy of calibration notions for ranking distributions (e.g., full-rank, rankwise, sub-k, and top-k calibration) and analyze their relationships systematically. This conceptual framework is one of the strongest contributions of the paper and provides a useful lens for understanding calibration behavior in ranking models.

S3. Strong theoretical component. The paper includes well-developed theoretical results characterizing implication and non-implication relationships among the proposed calibration notions. These results help clarify which calibration guarantees are stronger or weaker and provide a principled structure to the proposed taxonomy.

S4. Insightful empirical analysis. The experiments show that commonly used ranking models can exhibit significant calibration errors even when their ranking accuracy is strong. The application of the framework to RLHF reward models is particularly interesting and suggests that calibration captures aspects of model behavior that standard performance metrics may miss.

S5. Clear presentation and positioning. The paper is clearly written, the definitions are carefully motivated, and the overall narrative is easy to follow. The hierarchy of calibration notions and the associated figures help organize the conceptual contributions effectively.

There are also several opportunities for improvement that could further strengthen the work.  These include practical estimation of strong calibration notions, opportunities for approximate inference methods, and methods to improve (rather than only diagnose the lack of) calibration.  In other words, this paper opens up future research directions.

---

> ### Author Rebuttal · Authors · 2026-03-30
>
> We appreciate the positive assessment of our work and the suggestions for future directions.
> We are looking forward to addressing our outlined directions in future work and hope this work presents a first step to pursue this direction. We address your brief paragraph about weaknesses below.
>
> > [W1]: There are also several opportunities for improvement that could further strengthen the work. These include practical estimation of strong calibration notions, opportunities for approximate inference methods, and methods to improve (rather than only diagnose the lack of) calibration. In other words, this paper opens up future research directions.
>
> Thank you for your comments. Our work focuses on the development of calibration notions for the case of label ranking. Regarding practical estimation of strong calibration we refer to our answer to Q1 of reviewer atrk. Regarding the methods to improve calibration, we agree this is the most natural next step, and we discuss a potential strategy in our answer to Q4 of reviewer atrk.

---

> > ### Author Rebuttal · Reviewer_57eJ · 2026-04-01
> >
> > Thank you for your response!

---

### Decision · Program_Chairs · 2026-04-30

**Decision:**

Accept (regular)

**Comment:**

This work introduces new, meaningful notions of calibration when the outcome space consists of orderings of a label set. Reviewers appreciate the novelty, theoretical strength, empirical insight, and presentation quality. I recommend acceptance.